# Secondary metabolites of Hülle cells mediate protection of fungal reproductive and overwintering structures against fungivorous animals

**Li Liu[1], Christoph Sasse[1], Benedict Dirnberger[1], Oliver Valerius[1], Enikő Fekete-Szücs[1], Rebekka Harting[1], Daniela E Nordzieke[2], Stefanie Pöggeler[2], Petr Karlovsky[3], Jennifer Gerke[1]\*, Gerhard H Braus[1]\***

[1]University of Göttingen, Molecular Microbiology and Genetics and Göttingen Center for Molecular Biosciences (GZMB), Göttingen, Germany; [2]University of Göttingen, Genetics of Eukaryotic Microorganisms and Göttingen Center for Molecular Biosciences (GZMB), Göttingen, Germany; [3]University of Göttingen, Molecular Phytopathology and Mycotoxin Research, Göttingen, Germany

**Abstract** Fungal Hülle cells with nuclear storage and developmental backup functions are reminiscent of multipotent stem cells. In the soil, Hülle cells nurse the overwintering fruiting bodies of *Aspergillus nidulans*. The genome of *A. nidulans* harbors genes for the biosynthesis of xanthones. We show that enzymes and metabolites of this biosynthetic pathway accumulate in Hülle cells under the control of the regulatory velvet complex, which coordinates development and secondary metabolism. Deletion strains blocked in the conversion of anthraquinones to xanthones accumulate emodins and are delayed in maturation and growth of fruiting bodies. Emodin represses fruiting body and resting structure formation in other fungi. Xanthones are not required for sexual development but exert antifeedant effects on fungivorous animals such as springtails and woodlice. Our findings reveal a novel role of Hülle cells in establishing secure niches for *A. nidulans* by accumulating metabolites with antifeedant activity that protect reproductive structures from animal predators.

**\*For correspondence:**
jgerke@gwdg.de (JG);
gbraus@gwdg.de (GHB)

**Competing interest:** The authors declare that no competing interests exist.

## Introduction

Fungi are sessile organisms and cannot escape when they are attacked by predators or competitors. Whereas vertebrates have a protective immune system, which is regulated by Rel homology domain transcription factors, fungi have developed chemical defense strategies by producing protective secondary metabolites (SMs), many of which are regulated by the structurally similar velvet domain proteins (*Ahmed et al., 2013*). These small molecules (<1000 Da) are not directly involved in the growth of the producing organisms but play important roles in the organism's survivability in nature. Many fungal SMs affect the growth, survival, and reproduction of surrounding organisms and/or are toxic or deterrent to animals (*Künzler, 2018*; *Rohlfs and Churchill, 2011*). Former studies showed that the presence of bacteria or the predation by animals triggers fungal SM production (*Fischer et al., 2018*; *Volker Schroeckh and Brakhage, 2014*; *Xu et al., 2019*). Furthermore, secondary metabolism and development as well as primary metabolism are interconnected processes (*Bayram et al., 2016*; *Bayram et al., 2008*; *Sarikaya-Bayram et al., 2010*; *Keller, 2019*). Many fungal SMs possess intrinsic functions as components of developmental structures or signals initiating developmental processes. For example, fungal SMs as signaling hormones induce the formation of spores and regulate their

germination (*Niu et al., 2020*; *Rodríguez-Urra et al., 2012*). SMs required for the formation of fungal resting or sexual structures have been identified (*Calvo and Cary, 2015*; *Schindler and Nowrousian, 2014*; *Studt et al., 2012*).

The SMs epishamixanthone and shamixanthone were commonly isolated from several *Aspergillus* spp. (*Chen et al., 2016*). The cosmopolitan fungal genus *Aspergillus* comprises more than 300 species, some of which are used in biotechnology, cause human diseases, or spoil harvested crops (*Samson et al., 2014*). The biosynthetic pathway for epishamixanthone and shamixanthone was first identified in *Aspergillus nidulans* (*Sanchez et al., 2011*). The corresponding gene cluster consists of one polyketide synthase (PKS) encoding gene *mdpG* and 11 'tailoring' genes (*mdpA-F, mdpH-L*). The *mdp* genes are biosynthetically linked with the three *xpt* genes *xptA, xptB,* and *xptC*, which are scattered over the genome, and encode two prenyltransferases and one oxidoreductase (*Figure 1—figure supplement 1*). All biosynthetic genes will be referred to as the *mdp/xpt* gene cluster. In total, more than 30 compounds belonging mostly to anthraquinones, benzophenones, and xanthones are synthesized (*Caesar et al., 2020*; *Chiang et al., 2010*; *Pockrandt et al., 2012*; *Sanchez et al., 2011*). In *A. nidulans* wildtype A4, transcription of 10 out of 15 *mdp/xpt* genes was up-regulated under sexual development-inducing conditions and the three corresponding metabolites shamixanthone, emericellin, and emodin were detected in a metabolome analysis of cultures incubated under conditions inducing the sexual cycle (*Bayram et al., 2016*). SM-producing fungi possess global regulation mechanisms to control the SM production at specific times and in certain tissues for particular physiological functions (*Keller, 2015*). Whether the expression of the *mdp/xpt* cluster and the corresponding metabolites play a role in the sexual development of *A. nidulans* remains unknown.

*A. nidulans* is a soil-borne filamentous fungus with a well-characterized life cycle (*Park et al., 2019*). After spore germination, a network of vegetative hyphae is formed, which under certain environmental conditions matures through asexual or sexual developmental programs into spore-bearing conidiophores or sexual fruiting bodies (cleistothecia) (*Busch and Braus, 2007*; *Etxebeste et al., 2010*). Light stimulates the asexual pathway, whereas lowered oxygen levels and darkness stimulate the sexual pathway (*Bayram et al., 2016*). The cleistothecium serves as an overwintering structure and contains more than 10,000 sexual ascospores. It is surrounded by several layers of globose Hülle cells, which have nuclear storage and developmental backup functions and nurse the young fruiting body (*Troppens et al., 2020*). Lack of the epigenetic global regulator LaeA results in a loss of Hülle cells and in cleistothecia of reduced sizes (*Sarikaya-Bayram et al., 2010*), whereas lack of the velvet proteins VelB and VeA abolishes formation of cleistothecia and Hülle cells (*Bayram et al., 2008*; *Kim et al., 2002*).

Here, we investigated the localization of *mdp/xpt*-encoded proteins and their SM products during sexual development of *A. nidulans*. The Mdp/Xpt proteins are localized in sexual mycelia and Hülle cells, and the corresponding SMs are produced as soon as Hülle cells are present and cleistothecia begin to form. Furthermore, the loss of the regulatory velvet complex impaired the metabolite production by the *mdp/xpt* cluster. Strains with a disturbed biosynthetic pathway due to deletion of *mdp/xpt* genes could not produce the final products epi-/shamixanthone. Instead, they accumulated various intermediates in Hülle cells, leading to smaller Hülle cells with reduced activity and delayed maturation of cleistothecia. Food choice experiments showed that metabolites of the *mdp/xpt* cluster present in the wildtype protected *A. nidulans* from animal predators. These results suggest that xanthones produced by the *mdp/xpt* cluster in Hülle cells protect sexual fruiting body of *A. nidulans* from fungivorous animals.

## Results

### Proteins encoded by the *mdp/xpt* cluster are located in Hülle cells and sexual mycelia in *A. nidulans*

Most of the *mdp/xpt* genes in *A. nidulans* are expressed during sexual development (*Bayram et al., 2016*). A comparative proteome study on protein extracts of whole sexual tissues as well as enriched Hülle cells from wildtype A4 was conducted (*Figure 1—figure supplement 2* and Proteomic MS analysis data). Vegetative and asexual mycelia were used as controls. Vegetative mycelia were cultivated 20 hr in liquid medium and asexual and sexual tissues as well as Hülle cells were harvested 3, 5, and 7 days after inoculation on plates. An LC-MS analysis revealed that 24 proteins were present exclusively

**Table 1.** Five Mdp/Xpt proteins are found in sexual mycelia and Hülle cells.

| Gene ID (protein) | Putative function (Szwalbe et al., 2019) | 3 days | | 5 days | | 7 days | | 20 hr |
|---|---|---|---|---|---|---|---|---|
| | | Sexual mycelia | Hülle cells | Sexual mycelia | Hülle cells | Sexual mycelia | Hülle cells | Vegetative mycelia |
| AN10023 (MdpL) | Baeyer-Villiger monooxygenase | 121 | 25 | 98 | – | 34 | 42 | 9 |
| AN7998 (XptC) | Reductase | 54 | 7 | 48 | 6 | 26 | – | 1 |
| AN12402 (XptB) | Prenyltransferase | 52 | 5 | 19 | – | – | 8 | – |
| AN10022 (MdpH) | Anthrone oxidase, decarboxylase | 27 | – | 2 | 5 | 27 | 6 | – |
| AN0150 (MdpG) | Polyketide synthase | 12 | 4 | – | 3 | – | – | – |

in both sexual mycelia and enriched Hülle cells but were not identified from vegetative or asexual tissues (*Supplementary file 1*). Among them, five proteins encoded by the *mdp/xpt* cluster were identified, MdpG, MdpL, MdpH, XptB, and XptC (*Table 1*). To verify the localization of these proteins, the final enzyme in the biosynthesis of epi-/shamixanthone, XptC, was selected as an example and C-terminally fused to GFP for fluorescence microscopy. The fusion protein XptC-GFP was exclusively detected in 3-day-old sexual hyphae as well as in enriched Hülle cells but not in 20-hr-old vegetative hyphae (*Figure 1*). The stability of the fusion protein XptC-GFP was verified by applying an α-GFP antibody in Western analysis (*Figure 1b*). These results suggest that at least five members of the *mdp/xpt* cluster, MdpG, MdpL, MdpH, XptB, and XptC, are specifically localized to Hülle cells as well as sexual hyphae. Members of the Mdp/Xpt proteins can be detected from 3 to 7 days of sexual development.

## The *mdp/xpt* cluster metabolites change over time with fruiting body development

Under laboratory conditions, *A. nidulans* wildtype (AGB552), harboring Δ*nkuA* for improved homologous gene replacements, forms young sexual fruiting bodies (cleistothecia) that are surrounded by Hülle cells after 3 days of sexual growth in the absence of light. At this stage, all *mdp/xpt* genes are expressed (*Figure 2—figure supplement 1*). After 5 days, the cleistothecia are mature with a dark pigmented shell (*Figure 2—figure supplement 2*). Timing and localization of *mdp/xpt* metabolite production were monitored during sexual development and their roles in cleistothecia formation were examined. Different time points were selected for SM analysis (*Figure 2—figure supplement 2*): prior to the development of cleistothecia or Hülle cells (day 2), young cleistothecia with Hülle cells (day 3) and mature cleistothecia with Hülle cells (day 5), as well as late stages with mature cleistothecia (days 7 and 10). The different SM intermediates were addressed by a genetic approach. The pathway was disturbed by deleting *mdpG* and *mdpF* (early biosynthetic steps), *mdpC* and *mdpL* (intermediate biosynthetic steps), and *mdpD*, *xptA*, *xptB*, and *xptC* (late biosynthetic steps) separately (*Figure 2A*). *A. nidulans* wildtype and *mdp/xpt* deletion strains were grown under sexual conditions for 2, 3, 5, 7, and 10 days and the extra- and intracellular metabolites were extracted with ethyl acetate and subjected to LC-MS analysis. The identified *mdp/xpt* cluster products were categorized into four groups according to their molecular structures (anthraquinones in orange, benzophenones in blue, xanthones in green, arugosins in black, *Figure 2a*).

After 2 days of sexual development, wildtype and deletion strains did not produce any *mdp/xpt* metabolites (*Figure 2—figure supplement 3*). After 3 and 5 days, wildtype produced arugosin A (**1**) and the final xanthones emericellin (**2**), shamixanthone (**3**), and epishamixanthone (**4**) (*Figure 2b*, *Figure 2—figure supplement 3*, *Supplementary file 2*). As expected, loss of the first two enzymes of the biosynthesis MdpG and MdpF completely abolished the production of cluster metabolites. Deletion of the intermediate enzyme encoding genes *mdpC* and *mdpL* led to a loss of **1–4** but to the

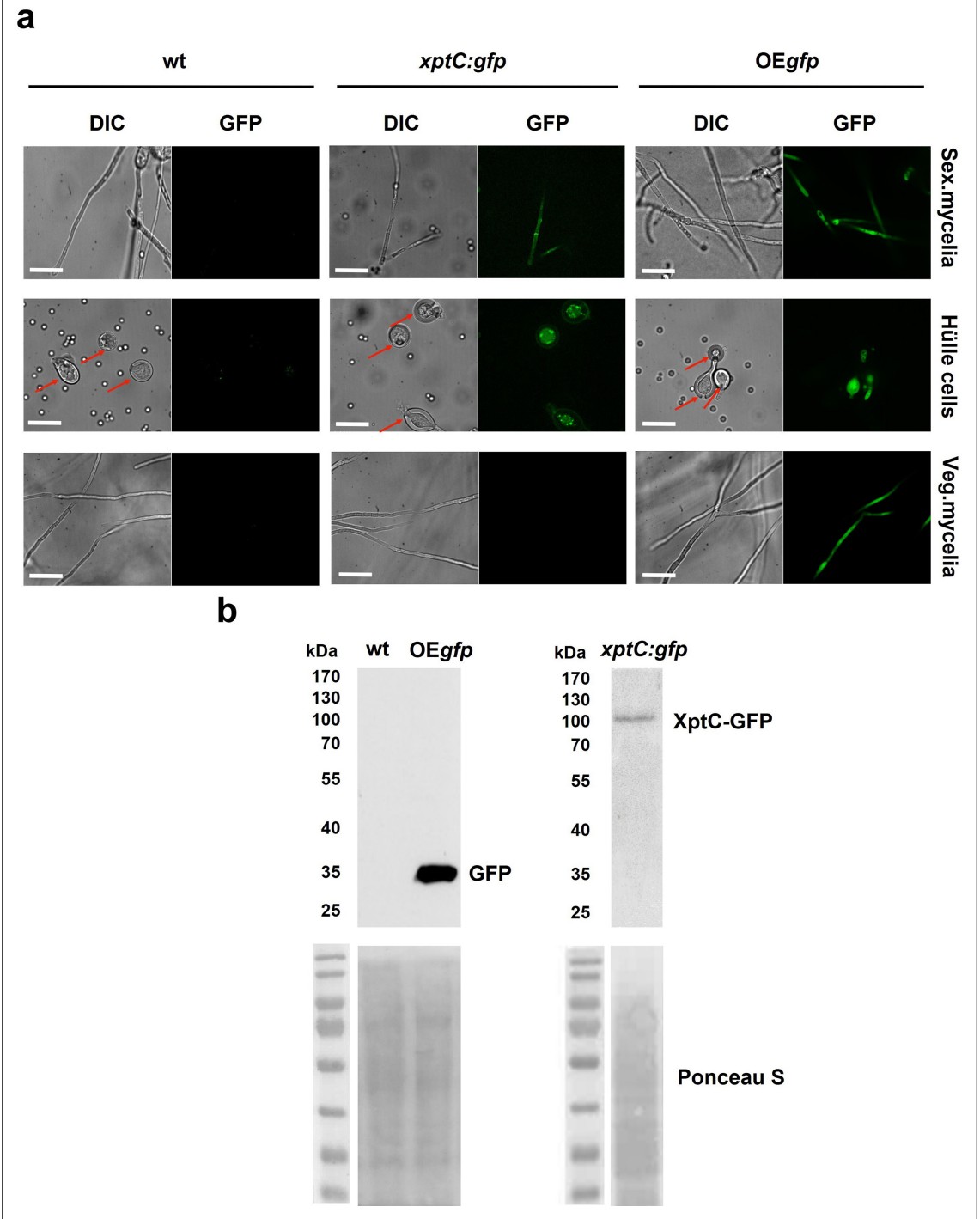

**Figure 1.** XptC is enriched in *Aspergillus nidulans* Hülle cells. (**a**) Fluorescence microscopy of sexual mycelia, Hülle cells (after 3 days of incubation), and vegetative mycelium after 20 hr of incubation from *xptC:gfp* strain. *A. nidulans* wildtype (wt, AGB552) and constitutively expressed GFP strain (OE*gfp*, AGB596) were used as controls. Red arrows indicate Hülle cells. Scale bar = 20 μm. The fusion protein XptC-GFP is undetectable in vegetative mycelia. (**b**) Western hybridization of sexual (sex) tissues harvested 3 days after inoculation. α-GFP antibody was used. The fusion protein XptC-GFP was detected at approximately 95 kDa. A strain expressing GFP constitutively (OE*gfp*, AGB596) and wildtype (wt, AGB552) served as controls. Ponceau S staining was used as sample loading control.

The online version of this article includes the following source data and figure supplement(s) for figure 1:

**Source data 1.** Raw data Western experiments.

**Source data 2.** Raw data Western experiments.

*Figure 1 continued on next page*

*Figure 1 continued*

**Source data 3.** Labelled, uncropped wWestern experiments.

**Figure supplement 1.** The epi-/shamixanthone-producing *mdp* and *xpt* genes are localized on different *Aspergillus nidulans* chromosomes.

**Figure supplement 2.** Mdp/Xpt proteins are expressed in sexual tissues of *Aspergillus nidulans*.

accumulation of the anthraquinones 2,$\omega$-dihydroxyemodin (**5**), $\omega$-hydroxyemodin (**6**) and emodin (**7**) in both strains and chrysophanol (**8**) in Δ*mdpL*. Deleting the biosynthetically final enzyme encoding genes *mdpD*, *xptA,* and *xptB* abolished the production of **1–4**. Δ*mdpD* and Δ*xptA* accumulate the xanthones paeciloxanthone (**9**) and variecoxanthone A (**10**), respectively. Δ*xptB* accumulates the anthraquinone **5**. In addition, these three strains accumulate some unidentified compounds. Deletion of *xptC* led to a loss of the final xanthones **3** and **4** but an increased accumulation of **2**. After 7 and 10 days of sexual development, the accumulated emodins **5–7** of the deletion strains were decreased or even disappeared (*Figure 2b*, *Figure 2—figure supplement 3*), whereas the abundance of **8** increased and the xanthones **2–4** and **10** were still detectable in similar amounts as after 3 and 5 days.

In summary, the SMs of the *mdp/xpt* cluster are produced as soon as the first sexual structures are visible after 3 days of sexual development. During cleistothecia maturation from 3 to 10 days, the amount of emodins gradually decreased, whereas the amount of chrysophanol and the xanthones increased or remained stable.

The medium from 3-day-old sexual mycelium of different strains was analyzed after removing the fungal colony to determine whether the identified metabolites were located exclusively within Hülle cells. LC-MS analysis revealed that the metabolites were also secreted to the medium, and thus may be secreted into the environment in natural habitats (*Figure 2—figure supplement 4*).

## The *mdp/xpt* cluster metabolites are enriched in Hülle cells of *A. nidulans*

The colors of the *mdp/xpt* cluster metabolites mostly are yellow or orange in pure powder form (*Chiang et al., 2010*). This enabled us to trace the localization of the metabolites in the fungus. *A. nidulans* wildtype forms a colony with green conidiospores and light yellow Hülle cells after 3 days of sexual growth (*Figure 3a and b*). All *mdp/xpt* deletion strains showed no change in green conidiospore production but the color of the Hülle cells changed. Whereas Δ*mdpD*, Δ*xptA*, Δ*xptB*, and Δ*xptC* showed no color difference to wildtype, the deletion strains of the early biosynthetic genes *mdpG* and *mdpF*, which have lost *mdp/xpt* metabolite synthesis (*Figure 2b*), produced colorless Hülle cells. Δ*mdpC*, which accumulates the brown (**5**) and yellow (**6**, **7**) emodins and yellow chrysophanol (**8**), produced dark yellow Hülle cells (*Figure 3a and b*). This indicates that the *mdp/xpt* metabolites are enriched inside the Hülle cells. The SMs have no obvious effect on Hülle cell shape (*Figure 3c*) but the accumulation of epi-/shamixanthone precursors in Δ*mdpC*, Δ*mdpL*, Δ*mdpD*, Δ*xptA*, and Δ*xptB* decreased the Hülle cell size (*Figure 3d*). The strains with the smallest Hülle cells, Δ*mdpC* and Δ*mdpL*, were tested for their Hülle cell germination ability. 39–56% of tested Hülle cells of wildtype and Δ*mdpG* germinated, whereas the small Hülle cells of Δ*mdpC* and Δ*mdpL* displayed a germination ability of only 2–6% (*Figure 3e*). The precursors accumulating in the deletion strains might inhibit the growth and germination of Hülle cells.

## The velvet complex regulates the *mdp/xpt* cluster metabolite production in *A. nidulans*

Velvet proteins, such as VeA and VelB, are fungal DNA-binding proteins with a similar structural fold as the mammalian NF-$\kappa$B inflammation and infection regulators (*Ahmed et al., 2013*). In *A. nidulans*, velvet proteins physically interact with epigenetic methyltransferases like LaeA (*Sarikaya-Bayram et al., 2014*; *Sarikaya-Bayram et al., 2015*). The heterotrimeric velvet complex VelB-VeA-LaeA coordinates sexual development and secondary metabolism (*Bayram et al., 2008*). VelB-VeA enters the nucleus to initiate sexual development and physically interacts with the epigenetic master regulator of secondary metabolism LaeA. To analyze the impact of the velvet complex on *mdp/xpt* SM production, extra- and intracellular metabolites of wildtype, Δ*veA*, Δ*velB*, and Δ*laeA,* were analyzed by LC-MS after 5 days of sexual development (*Figure 4*). The production of the final *mdp/xpt* products, arugosin A (**1**) and the xanthones emericellin (**2**), shamixanthone (**3**), and epishamixanthone (**4**), was abolished

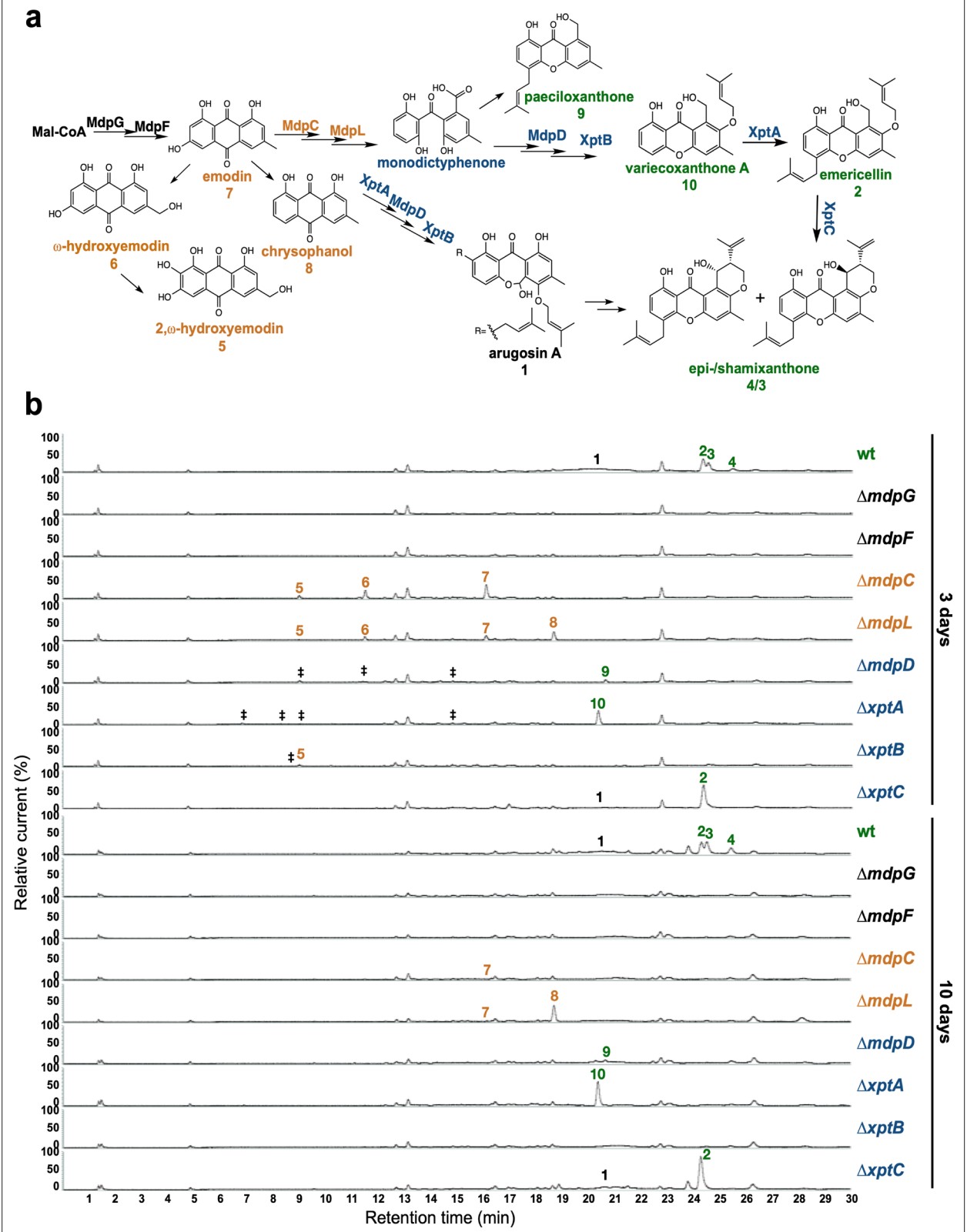

**Figure 2.** The *mdp/xpt* cluster metabolites are produced after 3 days of sexual growth in *Aspergillus nidulans*. (**a**) Simplified biosynthetic pathway of epi-/shamixanthone in *A. nidulans* (*Chiang et al., 2010*; *Pockrandt et al., 2012*; *Sanchez et al., 2011*). Black enzymes are localized at the early steps of the biosynthetic pathway. Orange enzymes are localized at the middle of the biosynthetic pathway. Blue enzymes are localized at the late steps of the biosynthetic pathway. The cluster products are classified as four groups: anthraquinones are orange, benzophenones are blue, xanthones are

*Figure 2 continued on next page*

*Figure 2 continued*

green, arugosin A is black. (**b**) Chromatograms of secondary metabolites (SMs) of *A. nidulans* wildtype (wt, AGB552) and *mdp/xpt* deletion strains (Δ). Conidia of *A. nidulans* wt and *mdp/xpt* mutant strains were point-inoculated on minimal medium (MM) and grown under conditions inducing the sexual cycle for 3 and 10 days. Extra- and intracellular metabolites were extracted and detected by LC-MS with a charged aerosol detector (CAD) in three independent experiments. Only CAD-detectable and identified SMs are shown with numbers: (**1**) arugosin A; (**2**) emericellin; (**3**) shamixanthone; (**4**) epishamixanthone; (**5**) 2,$\omega$-dihydroxyemodin; (**6**) $\omega$-hydroxyemodin; (**7**) emodin; (**8**) chrysophanol; (**9**) paeciloxanthone; (**10**) variecoxanthone A. ‡ marks unidentified compounds.

The online version of this article includes the following figure supplement(s) for figure 2:

**Figure supplement 1.** The *mdp/xpt* genes are expressed during fungal sexual development.

**Figure supplement 2.** *Aspergillus nidulans* developmental stages used for secondary metabolite analysis.

**Figure supplement 3.** Chromatograms of extracted secondary metabolites (SMs) from *Aspergillus nidulans* wildtype and *mdp/xpt* deletion strains.

**Figure supplement 4.** Chromatograms of secreted secondary metabolites (SMs) from *Aspergillus nidulans* wildtype and *mdp/xpt* deletion strains.

in Δ*veA* and Δ*velB* and was reduced in Δ*laeA*. This indicates that the velvet complex VelB-VeA-LaeA regulates the cluster metabolite production, whereby VelB and VeA are prerequisites for the *mdp/xpt* metabolite production.

## Accumulation of anthraquinone intermediates in *mdp/xpt* mutants impairs sexual development

Grown under conditions inducing the sexual cycle for 3 days, wildtype *A. nidulans* forms young, unpigmented cleistothecia covered with a high number of Hülle cells. Cleistothecia shells become pigmented after 4 days, and cleistothecia mature with a dark pigmented shell after 5 days (*Figure 5a*). The development of sexual structures was monitored over time in the wildtype and *mdp/xpt* deletion strains in order to reveal the role of metabolites encoded by the *mdp/xpt* cluster in sexual development. Deletion strains of *mdpG* and *mdpF*, which have not produced any *mdp/xpt* metabolites (*Figure 2b*), developed wildtype-like cleistothecia, indicating that xanthones and arugosin A produced by the *mdp/xpt* pathway are not required for sexual development. In contrast, Δ*mdpC* and Δ*mdpL*, which both accumulate the emodins **5–7**, were strongly delayed in cleistothecia development. After 3 days of sexual development, they formed pigmented Hülle cells without any cleistothecia. After 5 days, young, immature cleistothecia with soft and barely pigmented shells were formed. Cleistothecia were fully pigmented after 10 days (*Figure 5a*) but they exhibited reduced size (*Figure 5b*) and diminished number of ascospores (*Figure 5c*), which were viable (*Figure 5—figure supplement 1*). Even after 25 days, cleistothecia of Δ*mdpC* and Δ*mdpL* mutants were still significantly smaller than cleistothecia of the wildtype (*Figure 5—figure supplement 2*). The mutant Δ*mdpL* formed significantly more cleistothecia after 10 days as compared to all other strains (*Figure 5d*). The deletion strains of *mdpD*, *xptA*, and *xptB* exhibited a moderate delay of cleistothecia development (*Figure 5a*). After 4 days of sexual development, they formed young, only barely pigmented cleistothecia, but after 5 days the development was rescued and the cleistothecia turned wildtype-like. In addition, for Δ*xptB*, cleistothecia diameter (*Figure 5b* and *Figure 5—figure supplement 2*) as well as ascospore amount (*Figure 5c*) were diminished, but the spores were viable (*Figure 5—figure supplement 1*).

The effect of metabolites extracted from *mdp/xpt* deletion strains on cleistothecia development of *A. nidulans* wildtype was examined. Metabolites of wildtype and *mdp/xpt* deletion strains were extracted after 5 days of sexual development, dissolved in methanol, and loaded onto paper discs. SM-loaded paper discs were placed on an agar plate inoculated with *A. nidulans* wildtype spores. Pure methanol served as control. After 5 days of sexual incubation, the cleistothecia development on each paper disc was monitored. Metabolites of the wildtype and Δ*mdpG, ΔmdpF, ΔmdpL, ΔmdpD, ΔxptA, ΔxptB,* and Δ*xptC* displayed no effect on cleistothecia development. Metabolites of Δ*mdpC* exhibited marked effects on the development of wildtype cleistothecia, which were significantly smaller and remained immature (*Figure 5e* and *Figure 5—figure supplement 3*). This indicates that the metabolites of Δ*mdpC* 2,$\omega$-dihydroxyemodin (**5**), $\omega$-hydroxyemodin (**6**), and emodin (**7**) or at least one of them inhibited the development of cleistothecia. The wildtype AGB552 does not accumulate these intermediates under laboratory conditions, therefore it is unlikely that these SMs play any role in the ecology of *A. nidulans*. In summary, the disruption of the epi-/shamixanthone biosynthetic pathway causing the accumulation of intermediates, in particular emodins, resulted in a delay of

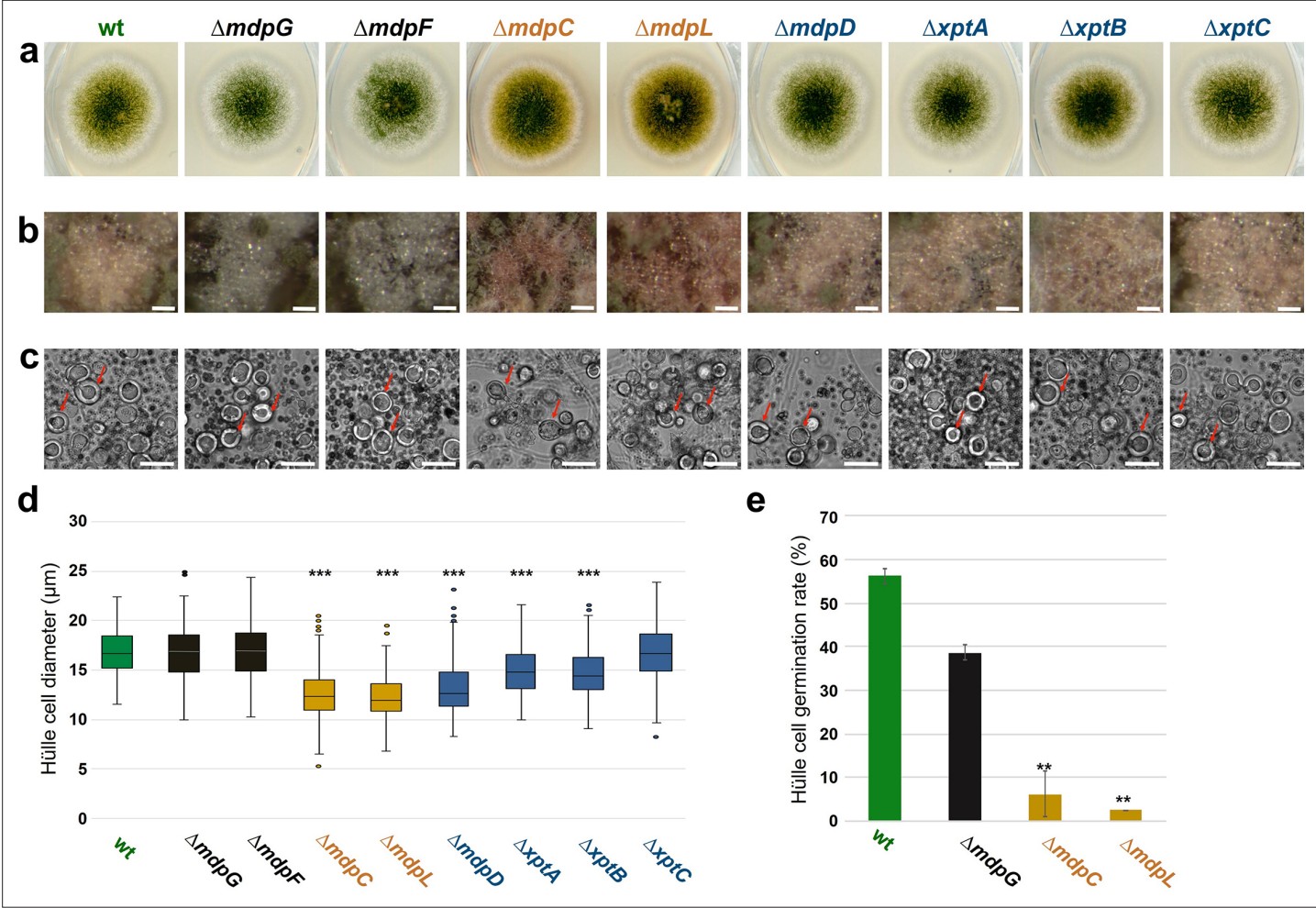

**Figure 3.** The *mdp/xpt* cluster metabolites are localized in Hülle cells. (**a**) Colony phenotypes of *Aspergillus nidulans* wildtype (wt, AGB552) and *mdp/xpt* deletion strains (Δ). Conidia were point-inoculated on minimal medium (MM) agar plates and cultivated 3 days under sexual conditions. (**b**) Photomicrographs of Hülle cells after 5 days. Scale bar = 50 μm. (**c**) Morphology of Hülle cells after 5 days. Red arrows indicate examples of Hülle cells. Scale bar = 25 μm. (**d**) Box plot of Hülle cells size after 5 days of sexual development (n ≥ 150). (**e**) Germination rate of Hülle cells. Detached Hülle cells were collected from cleistothecia surface after 5 days of sexual development and placed on fresh MM agar plates. The germination was monitored after 48 hr at 37 °C. n = 40 (±1) with two biological replicates. All significance tests are in comparison to wildtype (wt), \*\*\*/\*\*p < 0.005/0.05, two-tailed t-test.

The online version of this article includes the following figure supplement(s) for figure 3:

**Source data 1.** Hülle cells size after 5 days of sexual development.

**Source data 2.** Germination rate of Hülle cells.

cleistothecia maturation and a decrease in cleistothecial size. Another potential mechanism how the *mdp/xpt* pathway may affect sexual development is that it reduces the concentration of precursors, which might otherwise inhibit sexual development.

## Precursors of epi-/shamixanthone in *A. nidulans* but not the final products of the *mdp/xpt* pathway repress fruiting body and resting structure formation in other fungi

The soil fungus *A. nidulans* forms cleistothecia as overwintering structures to survive in harsh environments until its favorable growth conditions return. Epi-/shamixanthone precursors enriched in *mdp/xpt* deletion strains repressed cleistothecia development of *A. nidulans*. We analyzed the effect of the *mdp/xpt* metabolites on sexual reproduction or resting structures formation in other fungi. *Sordaria macrospora*, which lacks an asexual reproduction cycle, and two asexually reproducing *Verticillium* spp., which produce melanized resting structures, were investigated. SMs extracted from *A.*

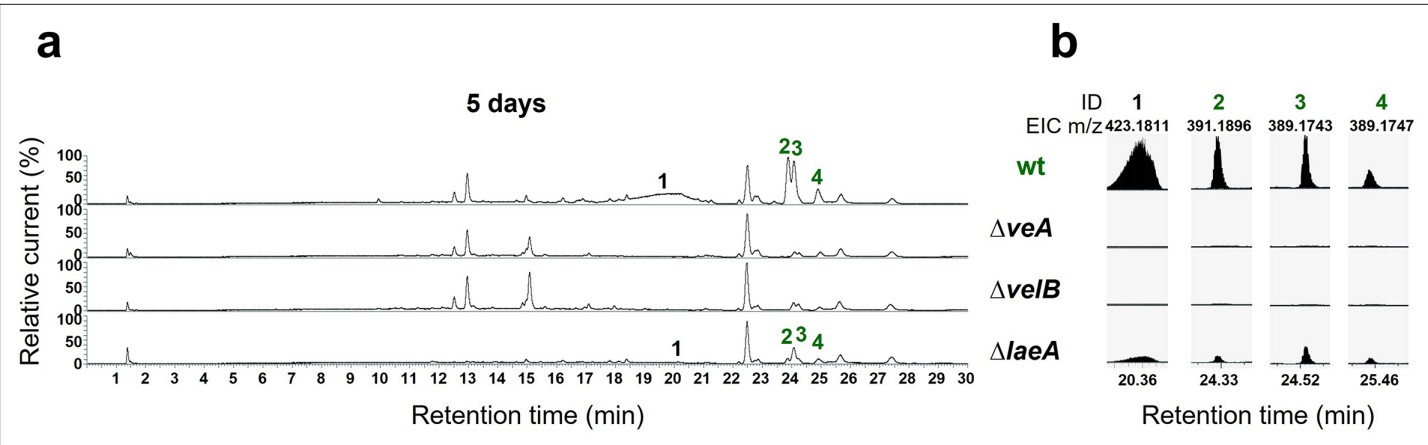

**Figure 4.** The velvet complex is required to produce *mdp/xpt* cluster metabolites. (**a**) Chromatograms of secondary metabolites (SMs) of *Aspergillus nidulans* wildtype (wt, AGB552) and velvet complex gene deletion strains (Δ). Conidia of *A. nidulans* strains were point-inoculated on minimal medium (MM) and sexually grown for 5 days. Extra- and intracellular metabolites were extracted and detected by LC-MS with a charged aerosol detector (CAD). Only CAD-detectable SMs of the *mdp/xpt* cluster are shown with numbers and were identified with MS and UV/VIS. (**b**) EICs (extracted ion chromatograms) of the compounds detected by CAD. *m/z* of **1** was detected in negative mode. *m/z* of **2**, **3**, and **4** was detected in positive mode. ID (compound number in this study): (**1**) arugosin A; (**2**) emericellin; (**3**) shamixanthone; (**4**) epishamixanthone.

nidulans wildtype and *mdp/xpt* deletion strains were loaded onto paper discs and placed on agar plates inoculated with fungal spores.

S. macrospora produces flask-shaped, pigmented sexual fruiting bodies (perithecia) after 7 days of surface cultivation (*Teichert et al., 2020*). Exposure of *S. macrospora* to metabolites secreted by the wildtype of *A. nidulans* did not affect perithecia formation. When *S. macrospora* was exposed to SMs of *A. nidulans* Δ*mdpC*, Δ*mdpL*, and Δ*mdpD*, perithecia formation was repressed, resulting in a halo surrounding the SM-loaded paper disc. The biggest halo was observed for Δ*mdpC* metabolites (*Figure 6*, *Figure 6—figure supplement 1a*). *Verticillium dahliae* and *Verticillium longisporum* form melanized hyphal aggregates called microsclerotia as resting structures to survive in the soil for decades (*Zeise and Tiedemann, 2001*). Exposed to the SMs of *A. nidulans* Δ*mdpC*, Δ*mdpL,* and Δ*mdpD*, *V. dahliae* and *V. longisporum* produced fewer microsclerotia under the paper discs and its surrounding area. Especially under the paper disc with SMs of Δ*mdpC*, *Verticillium* spp. cannot produce any microsclerotia (*Figure 6*). These results show that the precursors of epi-/shamixanthone accumulated by Δ*mdpC*, Δ*mdpL,* and Δ*mdpD* strains repressed sexual reproduction and resting structure formation in *S. macrospora* and *Verticillium* spp. The strongest effect was observed for Δ*mdpC*, which produced the emodins **5**, **6**, and **7**.

In order to identify the metabolites responsible for the effects, commercially available pure chemicals were tested; 75 µg of ω-hydroxyemodin (**6**), emodin (**7**), and chrysophanol (**8**) were loaded separately onto paper discs and placed on agar plates inoculated with *A. nidulans*, *S. macrospora*, *V. dahliae*, and *V. longisporum* wildtypes (*Figure 7*). 75 µg emodin corresponds to the amount extracted from approximately 6–7 point-inoculated *A. nidulans* Δ*mdpC* colonies after 3 days of growth. *S. macrospora*, *V. dahliae*, and *V. longisporum* produced fewer fruiting bodies or resting structures, respectively, when exposed to emodin (**7**) in comparison to the MeOH control, ω-hydroxyemodin (**6**), and chrysophanol (**8**). None of the metabolites affected the sexual fruiting body of *A. nidulans*. This effect seems to be specific to developmental structures because SMs produced by the Δ*mdpC* strain did not significantly affect the vegetative growth of *S. macrospora* hyphae (*Figure 6—figure supplement 1b*).

## The metabolite products of the *mdp/xpt* cluster in *A. nidulans* protect overwintering structures from animal predators

Besides competition with other microorganisms in soil, fungi face the risk of predation by fungivores. Therefore, we examined whether metabolites from the *mdp/xpt* cluster protected *A. nidulans* from predators. The *mdpC* and *mdpG* complementation strains (*mdpC* com, *mdpG* com), which produced

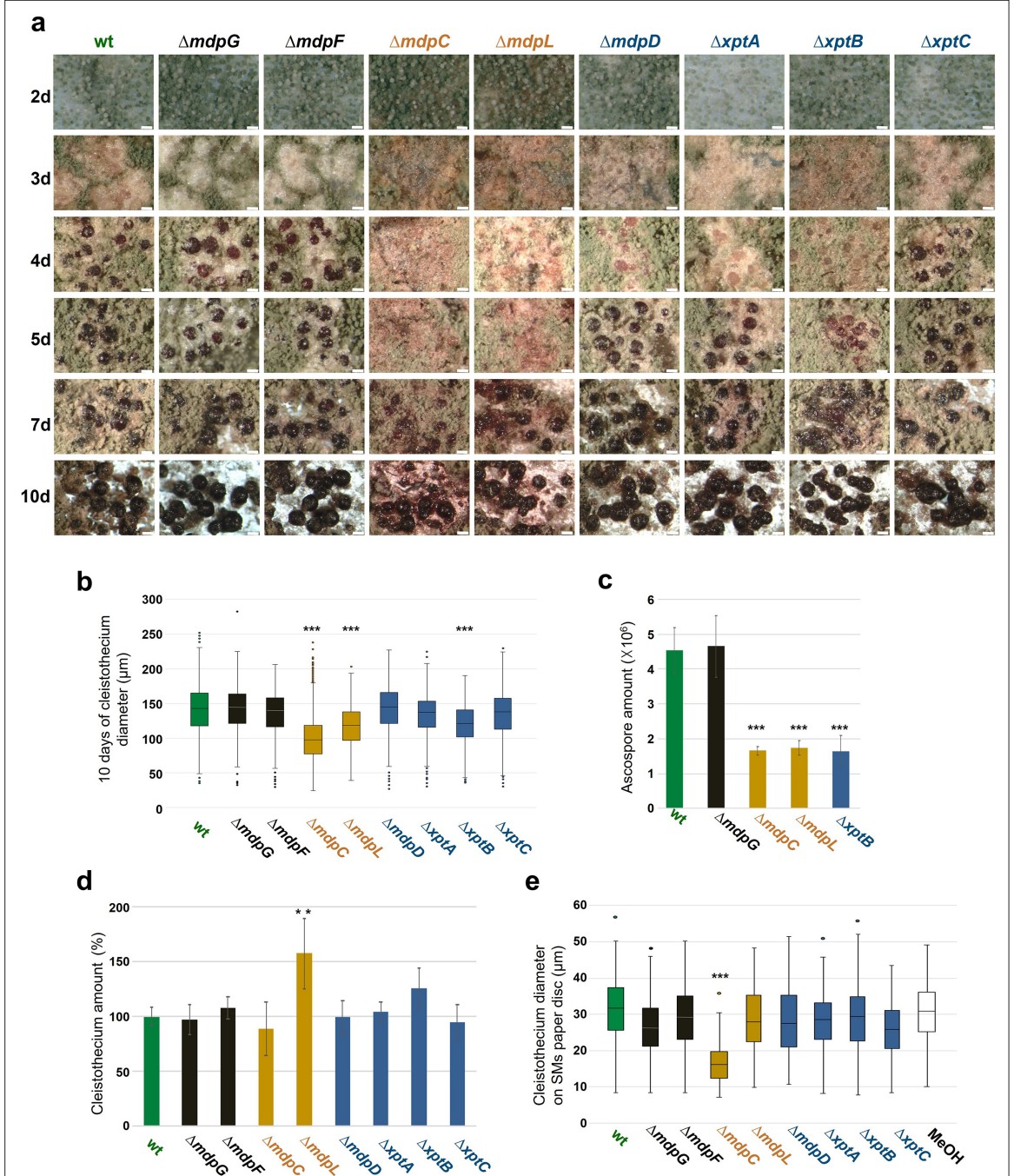

**Figure 5.** Intermediates of xanthone synthesis encoded by the *mdp/xpt* cluster repress sexual development of *Aspergillus nidulans*. (**a**) Photomicrographs of sexual structures of wildtype (wt, AGB552) and *mdp/xpt* deletion strains at different developmental stages. Scale bar = 100 μm. (**b**) Box plot of cleistothecia diameter after 10 days of sexual development (n ≥ 650). (**c**) Ascospore quantification after 10 days. Ten cleistothecia were broken in 100 μl 0.02 % Tween buffer and released ascospores were quantified. Error bar indicates standard deviation with three biological and three technical replicates. (**d**) Cleistothecia quantification after 10 days. Error bar indicates standard deviation with five biological replicates; amount of cleistothecia of wt was set to 100 %. (**e**) Box plot of cleistothecia diameter of *A. nidulans* wt grown on paper discs loaded with extracted secondary metabolites (SMs) of wt and *mdp/xpt* deletion strains. Strains were sexually grown for 5 days. SMs were extracted, solved in MeOH, and loaded on paper discs separately (pure MeOH was used as blank control). Paper discs were placed on agar plates inoculated with 1 × 10^5 conidia of *A. nidulans* wt. Cleistothecia on the paper discs were collected after 5 days of sexual growth (n ≥ 65). All significance tests are in comparison to wt, ***/**p < 0.005/0.05, two-tailed t-test.

The online version of this article includes the following figure supplement(s) for figure 5:

*Figure 5 continued on next page*

*Figure 5 continued*

**Source data 1.** Cleistothecia diameter after 10 days of sexual development.

**Source data 2.** Ascospore quantification after 10 days.

**Source data 3.** Cleistothecia quantification after 10 days.

**Source data 4.** Cleistothecia diameter of *Aspergillus nidulans* wildtype grown on paper discs loaded with *mdp/xpt* deletion strain secondary metabolites (SMs).

**Figure supplement 1.** Ascospores from small cleistothecia of Δ*mdpC* and Δ*xptB* are viable.

**Figure supplement 2.** Box plot of cleistothecia size after 25 days of sexual growth.

**Figure supplement 2—source data 1.** Cleistothecia size after 25 days of sexual growth.

**Figure supplement 3.** Secondary metabolites (SMs) of Δ*mdpC* affect the cleistothecia development of *Aspergillus nidulans* wildtype.

all *mdp/xpt* metabolites as the wildtype (***Figure 8—figure supplement 1***), the non-producing strain Δ*mdpG* as well as the emodins-accumulating strain Δ*mdpC* were offered to animal predators in a food choice experiment (***Figure 8—figure supplement 2***). Animals representing distant arthropod lineages were selected: the mealworm larvae *Tenebrio molitor* (insect), the collembolan *Folsomia candida* (primitive arthropod), and the woodlouse *Trichorhina tomentosa* (crustacean). Two fungal cultures were placed onto opposite sides of a Petri dish, and active animals were placed onto the center area. The animals feeding on each fungal culture were counted at time intervals.

*T. molitor* showed no obvious food preference for *mdp/xpt* metabolites (***Figure 8—figure supplement 2a***). *F. candida* and *T. tomentosa* strongly avoided *A. nidulans* producing the final products

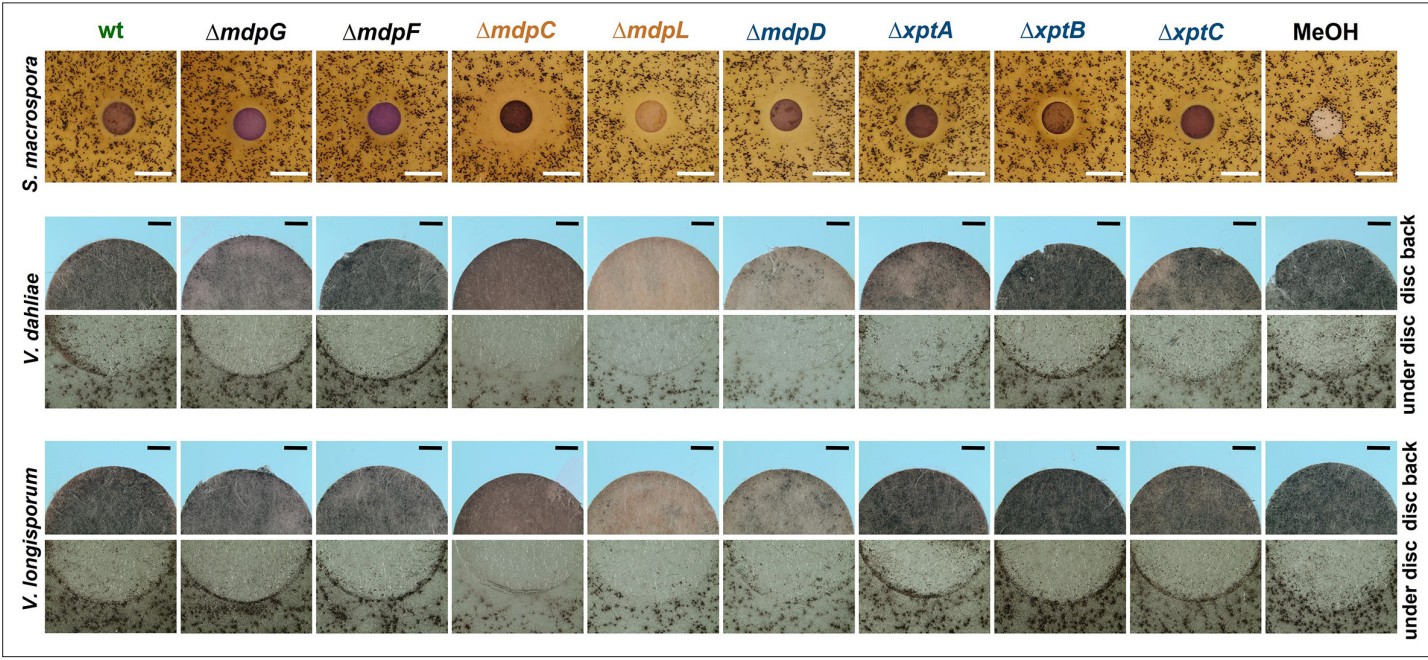

**Figure 6.** Intermediates of the *mdp/xpt* cluster repress fungal reproduction and resting structure formation. Microphotographs of fungal reproduction and resting structures exposed to extracted secondary metabolites (SMs) of *Aspergillus nidulans* wildtype (wt) and *mdp/xpt* deletion strains. Conidia of *A. nidulans* strains were point-inoculated on minimal medium (MM) and sexually grown for 5 days. SMs were extracted, solved in MeOH, and loaded onto paper discs (MeOH solvent served as control). Paper discs were placed on agar plates inoculated with spores of the tested fungi. For *Sordaria macrospora*, BMM agar plates were inoculated with $2 \times 10^5$ spores and cultivated 7 days at 27 °C. White bar = 1 cm. For *Verticillium* spp., simulated xylem medium (SXM) agar plates were inoculated with $1 \times 10^5$ *Verticillium dahliae* or *Verticillium longisporum* spores and cultivated 10 days at 25 °C. The upper panel shows pictures taken from the back of the paper disc and the lower panel shows the agar under the paper disc. Black bar = 1 mm.

The online version of this article includes the following figure supplement(s) for figure 6:

**Figure supplement 1.** Effect of secondary metabolites (SMs) on perithecium formation and vegetative growth of *Sordaria macrospora*.

**Figure supplement 1—source data 1.** Effect of secondary metabolites on perithecium formation.

**Figure supplement 1—source data 2.** Effect of secondary metabolites on vegetative growth of *Sordaria macrospora*.

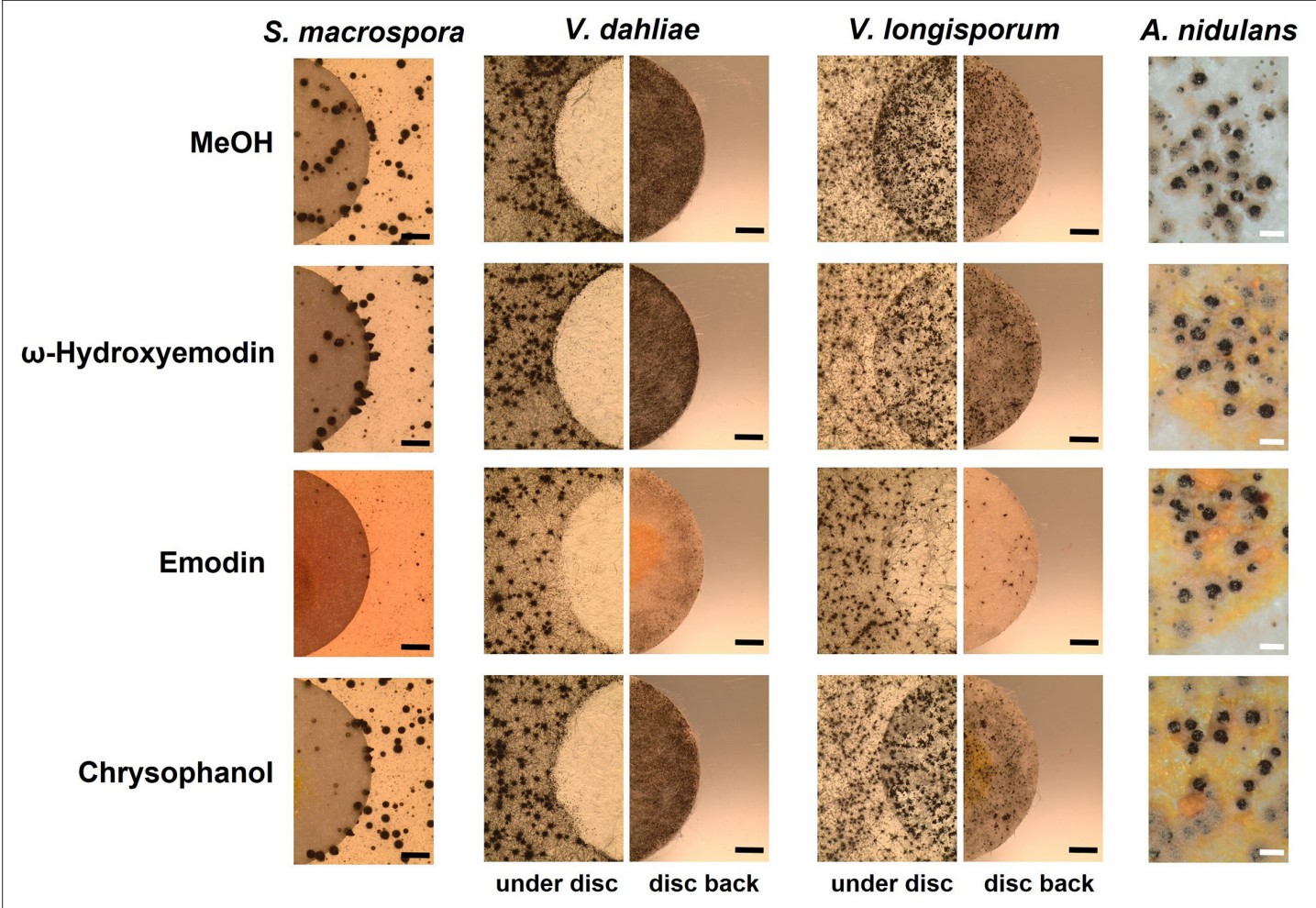

**Figure 7.** Emodin represses fungal reproduction and resting structure formation. Pure emodin, ω-hydroxyemodin, and chrysophanol were dissolved in MeOH and loaded onto paper discs (final amount 75 µg per disc). MeOH served as blank control. Paper discs were placed on agar plates inoculated with spores of the tested fungi. For *Sordaria macrospora*, BMM agar plates were inoculated with $2 \times 10^5$ spores and cultivated 7 days at 27 °C. For *Verticillium* spp., simulated xylem medium (SXM) agar plates were inoculated with $1 \times 10^5$ spores and cultivated 10 days at 25 °C. Microsclerotia were monitored under the paper disc and at its back. Black bar = 1 mm. For *Aspergillus nidulans*, $1 \times 10^5$ conidia of wildtype were inoculated on minimal medium (MM) agar plates and sexually incubated for 5 days at 37 °C. White bar = 200 µm.

including epi-/shamixanthone (*Figure 8—figure supplement 2b,c*). The springtails and isopods gradually gathered on fungal cultures not producing the final products, where they remained over 24 hr, indicating that the final products deterred the animals from feeding. Emodins, which accumulated in Δ*mdpC*, have not affected animal grazing. A similar number of animals were found on Δ*mdpC* as on Δ*mdpG* cultures. In an additional experiment, the feeding behavior of the animals was analyzed. *F. candida* and *T. tomentosa* were placed on agar pieces with 5-day-old sexual mycelia of the wildtype, the Δ*mdpG*, and the Δ*mdpC* strain. Fungal cultures were incubated for 6 days with the predators. Only cultures of the mutant strains showed large areas of mycelium consumed by both animal species (*Figure 8a–b*), indicating that the animals preferred feeding on mycelium without *mdp/xpt* xanthones. This result raised the question whether the observed protection was limited to the mycelium or also included the cleistothecia. The question was addressed by exposing 4-day-old cleistothecia with Hülle cells of the wildtype and the Δ*mdpG* strain to *T. tomentosa* and *F. candida* on agar plates. The *mdpC* deletion strain could not be included because of its incapability to produce cleistothecia in 4 days. Only a few cleistothecia were consumed by *T. tomentosa* (less than 10 % for the wildtype and Δ*mdpG*), regardless of the strain. No differences in the feeding behavior of *T. tomentosa* on cleistothecia of the wildtype and the Δ*mdpG* strain were observed. In contrast, *F. candida* showed a significant feeding preference for cleistothecia of the Δ*mdpG* strain in comparison to the wildtype (*Figure 8c*).

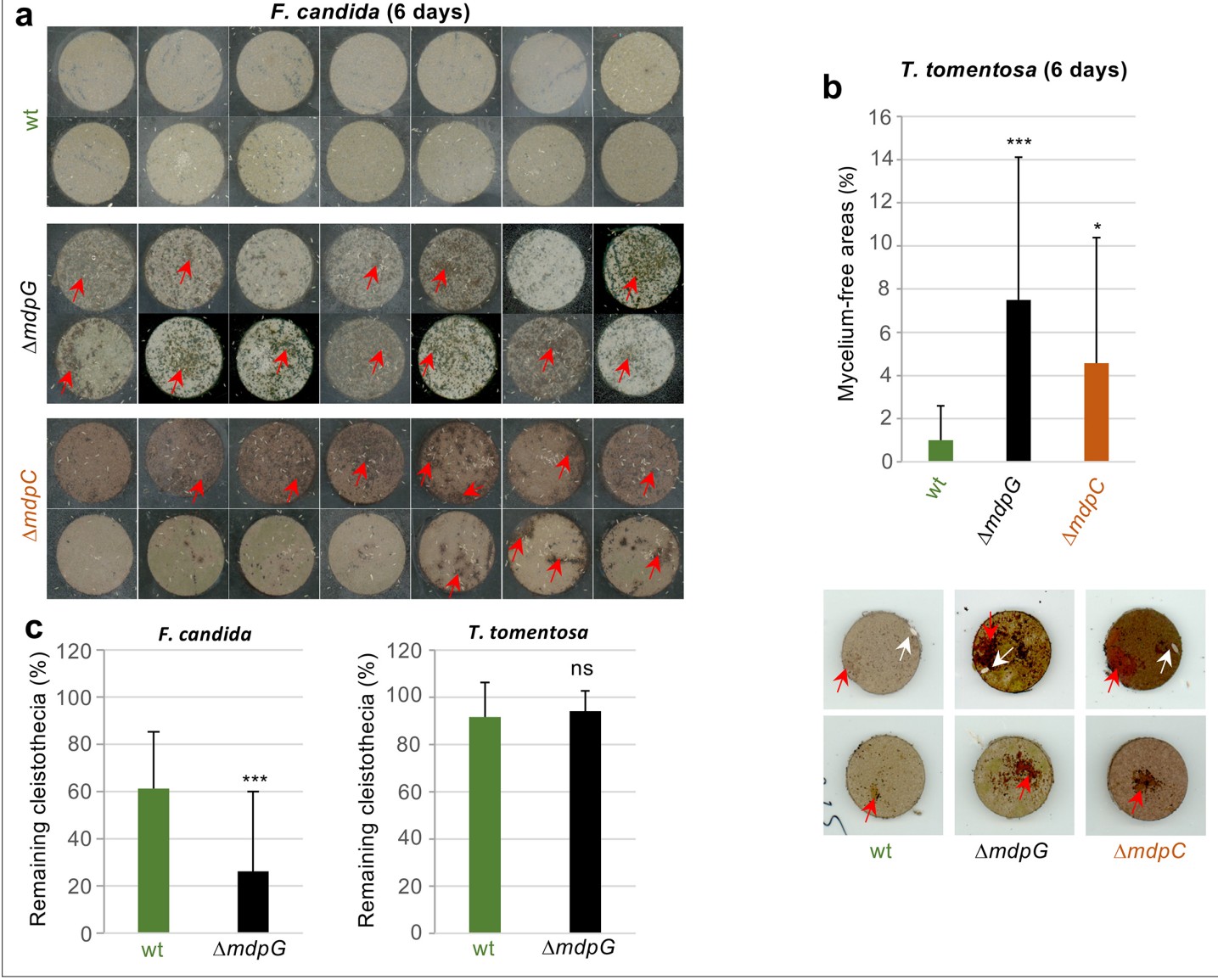

**Figure 8.** Metabolites of the *mdp/xpt* cluster in *Aspergillus nidulans* protect cleistothecia and sexual mycelium from fungal predators. Approximately 30 springtail animals of *Folsomia candida* were placed on agar pieces covered with mycelium of fungal strains induced for 5 days under sexual conditions. Agar pieces with animals were incubated for 6 days at 22 °C. For the two *A. nidulans* mutant strains ΔmdpG and ΔmdpC, mycelium-free areas could be observed (areas are indicated by red arrows). None of the agar pieces with mycelium of the wildtype showed large areas without mycelium. Two independent experiments were carried out. Each experiment contains at least six agar pieces from two different plates. (**b**) The same experimental setup as in (**a**) was used with the isopod *Trichorhina tomentosa*. Five animals were put on each agar piece. After 6 days, mycelium-free areas larger than 1 % of the area of the agar piece were quantified using ImageJ software. Red arrows show mycelium-free areas. White arrows point on single *T. tomentosa* animals on the agar pieces. (**c**) Ten cleistothecia with Hülle cells of the indicated strains obtained from mycelium incubated for 4 days under sexual-inducing conditions were placed on water-agar plates and incubated with *T. tomentosa* (five animals) or *F. candida* (approximately 30 animals). Cleistothecia remaining after 24 hr were counted. Two biological replicates with *T. tomentosa* and four biological replicates with *F. candida* were performed. Each biological replicate contained six technical replicates. Statistical significance was determined using two-tailed t-test with *p < 0.05, ***p < 0.005 (referenced to wt). Error bar indicates the standard deviation.

The online version of this article includes the following figure supplement(s) for figure 8:

**Source data 1.** Eaten areas of sexual mycelium by *Trichorhina tomentosa* after 6 days.

**Source data 2.** Eaten cleistothecia by the predators *Trichorhina tomentosa* and *Folsomia candida*.

**Figure supplement 1.** Complementation strains of *mdpG* and *mdpC* restored the phenotype and the secondary metabolite (SM) production.

**Figure supplement 2.** The final secondary metabolite (SM) products of the *mdp/xpt* cluster repel fungal predators.

*Figure 8 continued on next page*

*Figure 8 continued*

**Figure supplement 2—source data 1.** The final secondary metabolite (SM) products of the *mdp/xpt* cluster repel fungal predators.

**Figure supplement 3.** Final metabolite products of the *mdp/xpt* cluster in *Aspergillus nidulans* wildtype have no toxic effect on fungal predators.

**Figure supplement 3—source data 1.** Toxicity of secondary metabolites to *Trichorhina tomentosa* and *Folsomia candida*.

The results showed that metabolites produced by the *mdp/xpt* pathway protected sexual mycelium as well as cleistothecia from predators.

Toxicity of the metabolites emericellin, shamixanthone, or epishamixanthone to arthropods was examined as potential cause for feeding suppression of animal predators on cultures of the wildtype strain. *F. candida* and *T. tomentosa* were fed with metabolite extracts of the wildtype, Δ*mdpG* or Δ*mdpC* strains. Mortality rates in *T. tomentosa* and *F. candida* fed on metabolites of the wildtype and mutant strains were indistinguishable, showing that xanthones produced by the *mdp/xpt* pathway are not toxic to animals (**Figure 8—figure supplement 3**). Taken together, metabolites produced by the *mdp/xpt* pathway in *A. nidulans* are nontoxic antifeedants preventing predators *F. candida* and *T. tomentosa* from grazing.

## Discussion

SMs (syn. specialized metabolites) are low-molecular-weight compounds with a high structural diversity enabling different biological functions. Many SMs possess protective properties for the producing organism and are synthesized in response to abiotic and biotic stimuli. They protect fungi against environmental stresses such as oxidation and UV radiation (**Rao et al., 2017**) and inhibit bacterial or fungal competitors (**Künzler, 2018**; **Stroe et al., 2020**; **Wheatley, 2002**). Besides, many SMs exert antifeedant activities. Aurofusarin protects *Fusarium* species from animal grazing and asparasone A protects *Aspergillus flavus* from insect predators (**Cary et al., 2014**; **Xu et al., 2019**). The main finding of this study is that the soil fungus *A. nidulans* accumulates various xanthones in a specific fungal sexual cell type, the Hülle cells, and that these xanthones protect its sexual structures from predation by exerting antifeedant effects on fungivorous animals.

Fruiting bodies (cleistothecia) of *A. nidulans* serve as reproductive and overwintering structures, and their formation is highly material- and energy-consuming (**Pöggeler et al., 2006**). During overwintering, cleistothecia secure fungal survival. Thus, a potent protection mechanism against predators is needed to ensure long-term survival. While feeding on *A. nidulans,* the fungivorous springtail *F. candida* prefers vegetative hyphae and conidia to cleistothecia (**Döll et al., 2013**). The cleistothecia are covered with globose Hülle cells which are suggested to serve as nursing cells (**Sarikaya-Bayram et al., 2010**) and fungal backup stem cells with nuclear storage function to produce genetically diverse spores in changing nutrient conditions (**Troppens et al., 2020**). Here, we showed that xanthone metabolites of the *mdp/xpt* pathway are produced and accumulate in Hülle cells (**Figure 3**), and that they deter animal predators from feeding on the fungus (**Figure 8**, **Figure 8—figure supplement 2**). This suggests an additional role for Hülle cells to secure survival of cleistothecia. The *mdp/xpt* metabolites are produced as soon as the first Hülle cells appear, protecting sexual mycelium and the developing, young fruiting body. They are still present when the cleistothecia mature, guaranteeing long-term protection (**Figure 2b**, **Figure 2—figure supplement 3**, and **Figure 5a**). It will be an interesting question whether there are also conditions where *A. nidulans* is capable of inducing the *mdp/xpt* cluster in the presence of specific animal fungivors to protect its mycelium. Fruit fly larvae preferred grazing on strains deleted for the regulatory *veA* or *laeA* genes (**Trienens et al., 2010**; **Trienens and Rohlfs, 2012**). Both encoded proteins are components of the trimeric velvet complex, which is involved in the coordinated regulation of secondary metabolism and sexual development. In *A. nidulans*, deletion of either gene resulted in defective Hülle cell and fruiting body development and in disturbed SM production (**Bayram et al., 2008**; **Sarikaya-Bayram et al., 2010**). Transcriptome analysis revealed that VeA positively regulates the expression of 8 out of 15 *mdp/xpt* genes (**Lind et al., 2015**). Here, deletion of the genes for VeA and VelB abolished the production of *mdp/xpt* metabolites (**Figure 4**). Deletion of *laeA* reduced the number of Hülle cells as well as the production of *mdp/xpt* metabolites. These findings indicate that the production of the antifeedant *mdp/xpt* metabolites is under the regulatory control of the velvet complex.

The *mdp/xpt* gene cluster is moderately conserved in two closely related *Aspergillus* species, *Aspergillus versicolor* and *Aspergillus sydowii* (*de Vries et al., 2017*). *A. nidulans* and *A. versicolor* form cleistothecia, whereas no sexual cycle has been found yet for *A. sydowii* (*Raper and Fennell, 1965*), though it forms Hülle cells (*Dyer and O'gorman, 2012*). The velvet complex VelB-VeA-LaeA is conserved in the fungal kingdom (*Bayram and Braus, 2012b*) and it is required for Hülle cell formation (*Sarikaya-Bayram et al., 2010*; *Kim et al., 2002*). The strains Δ*veA* and Δ*velB* cannot produce Hülle cells and Δ*laeA* just produces a few Hülle cells. This is well connected with the *mdp/xpt* metabolites production in velvet complex gene deletion strains (*Figure 4*) and suggests that the velvet complex regulation of the *mdp/xpt* cluster is Hülle cell-dependent. The *Aspergillus* section *Nidulantes* is subdivided into 7 clades and 65 species. Three clades and 34 species have a sexual state forming the fruiting body cleistothecium, typically surrounded by layers of Hülle cells. Therein, 27 species have been identified to produce *mdp/xpt* metabolites, for example, emericellin, 2, $\omega$ -hydroxyemodin or shamixanthone (*Chen et al., 2016*). The correspondence between *Aspergillus* species producing Hülle cells and those possessing *mdp/xpt* clusters raises the question whether Hülle cell-associated metabolites protect reproduction structures in all these species.

The *mdp* gene cluster is biosynthetically connected with three *xpt* genes, which are scattered over the genome. The *mdp* genes produce emodin as well as monodictyphenone, whereas the *xpt* genes use the *mdp* metabolites as precursors, converting them to xanthones and arugosin A (*Figure 2a*; *Chiang et al., 2010*; *Pockrandt et al., 2012*; *Sanchez et al., 2011*). Emodin can inhibit the formation of fruiting bodies and resting structures of other fungi such as *S. macrospora* and two different species of *Verticillium*, but not of *A. nidulans* itself (*Figure 7*). This might be explained by the fact that *A. nidulans* is able to convert emodin to other structures, whereas *S. macrospora* and *Verticillium* spp. do not possess enzymes for the conversion. This kind of detoxification is one common self-protection mechanism of fungi (*Keller, 2015*). The fact that emodin represses the formation of fruiting and resting structures in such divergent fungi as *S. macrospora* and *Verticillium* spp. suggests a commonly present emodin target. Emodin is widely identified in plant families but also in fungi, particularly in *Penicillium* spp. and *Aspergillus* spp. It exhibits a wide spectrum of ecological functions and protects higher plants against herbivores, pathogens, competitors, as well as extrinsic abiotic factors (*Izhaki, 2002*). In addition, emodin has pharmaceutical properties, such as anti-inflammatory and anti-tumor activities in mammalian cells by affecting the cellular NF- $\kappa$ B and MAPK (mitogen-activated protein kinase) signaling pathways (*Huang et al., 2004*; *Wang et al., 2006*; *Xie et al., 2019*). The MAPK signaling pathways are conserved from yeast to human playing a central role in transducing extracellular stimuli into intracellular responses (*Chen and Thorner, 2007*; *Meskiene and Hirt, 2000*). It is well known that MAPK pathways are commonly involved in fungal development (*Bayram et al., 2012a*). The fungal MAPK signaling pathways could be a suitable emodin target, disordering the signaling transduction and inhibiting the sexual fruiting body or resting structure formation. In some fungi, emodin might act as an agent of interference competition, securing space and nutrients for biomass accumulation and sexual fruiting body development. In our study, the analyzed *A. nidulans* AGB552 wildtype strain did not produce emodin under tested laboratory conditions, therefore it is unlikely that emodin has a direct ecological function. However, emodin production has been described in the *A. nidulans* natural isolate A4 (*Bayram et al., 2016*).

It is conceivable that the *mdp/xpt* pathway can produce emodin rather than terminal xanthones under certain conditions in natural habitats. The mechanism could include a selective, limited inactivation of the velvet complex by disassembly or protein stability control or action of other regulators of the *mdp/xpt* pathway. The scenario is however highly speculative. We rather suggest that the synthesis of antifeedant xanthones in *A. nidulans* evolved from a pathway producing interference competition metabolites in an ancestral species.

Besides emodin, other *mdp/xpt* SMs seem to be involved in the repression of fruiting and resting structure formation, since SMs of Δ*mdpD*, which do not contain emodin, are involved in repression in *S. macrospora*, *V. dahliae*, and *V. longisporum* (*Figure 6*). Further, enriched intermediates in the *mdp/xpt* deletion strains Δ*mdpC*, Δ*mdpL*, Δ*mdpD*, Δ*xptA*, and Δ*xptB* delayed the fruiting body maturation of *A. nidulans* (*Figure 5a*). Whereas the delay was gradually rescued along with the decrease of accumulated intermediates in Δ*mdpL*, Δ*mdpD*, Δ*xptA*, and Δ*xptB* (*Figure 2B* and *Figure 2—figure supplement 3*), Δ*mdpC* showed an additional reduction in cleistothecia size even after 25 days (*Figure 5—figure supplement 2*). Emodin and $\omega$ -hydroxyemodin do not show an effect on cleistothecia size when

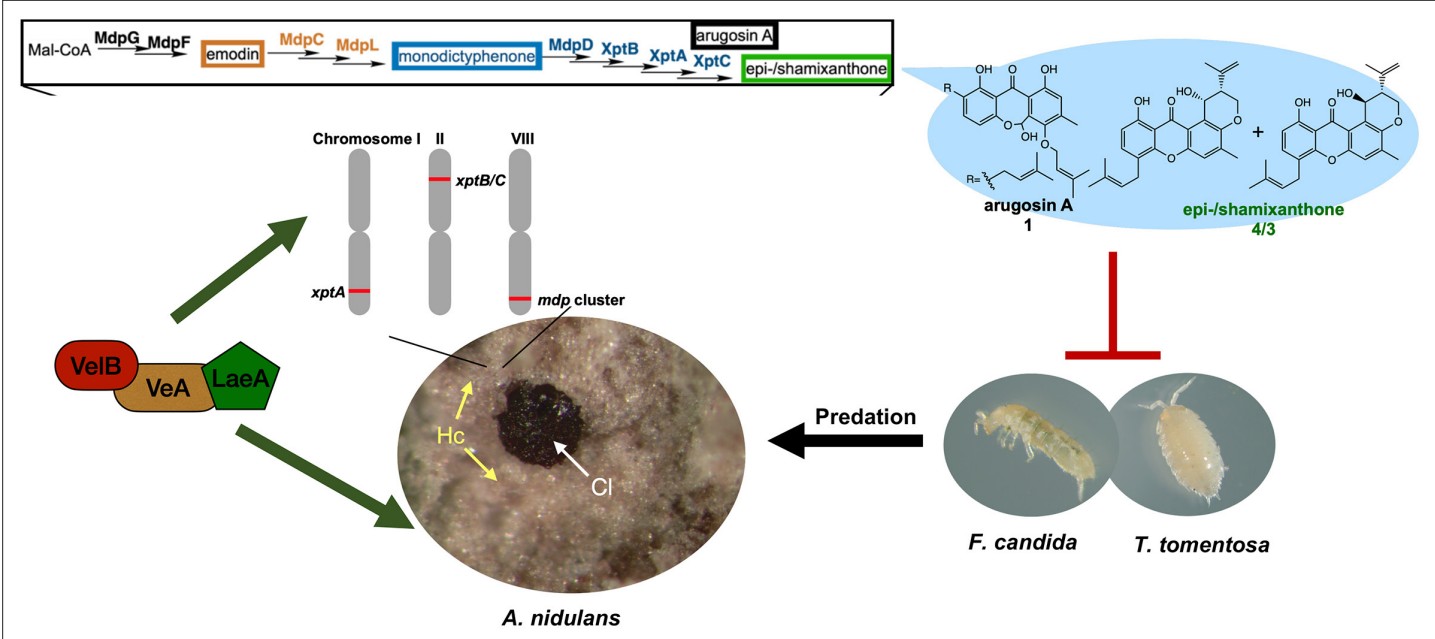

**Figure 9.** The *mdp/xpt* cluster metabolites establish a secure niche for *Aspergillus nidulans*. The velvet complex VelB-VeA-LaeA regulates sexual development of *A. nidulans* and the expression of *mdp/xpt* genes. The resulting metabolites are accumulated in the sexual fruiting body (Cl) nursing Hülle cells (Hc). The final products arugosin A (1) and epi-/shamixanthone (4/3) protect sexual fruiting bodies from the animal predators *Folsomia candida* and *Trichorhina tomentosa*.

externally added to *A. nidulans* wildtype (**Figure 7**). Therefore, 2,$\omega$-dihydroxyemodin or a combination of the different anthraquinone SMs might be the responsible compounds for the reduction in cleistothecia size. Smaller cleistothecia have lower energy and material costs. The accumulation of those intermediates might be an internal signal for incomplete xanthone/arugosin biosynthesis, which leads to fruiting bodies unprotected from animal predators. SMs produced by the Δ*mdpC A. nidulans* strain reduce fruiting body formation, but do not affect vegetative growth of *S. macrospora* (**Figure 6—figure supplement 1b**). Moreover, they had no toxic effects on the tested fungal predators (**Figure 8—figure supplement 3**). Therefore, it seems that the intermediates as well as the final products of the *mdp/xpt* cluster have rather a limited regulatory effect on the developmental program than a general toxic inhibiting influence on the fungus.

In conclusion, our results suggest that the *mdp/xpt* pathway in *A. nidulans* was recruited for the protection of fruiting bodies from predation. This occurred by (i) connecting the anthraquinone producing *mdp* cluster with *xpt* genes converting anthraquinones into xanthones with antifeedant activity; (ii) expressing the *mdp/xpt* pathway in Hülle cells that surround the fruiting bodies; and (iii) coordinating *mdp/xpt* expression with sexual development by the velvet complex. These findings support a new role of Hülle cells as protective cells. They establish a secure niche for *A. nidulans* by accumulating metabolites with antifeedant activity that protect the reproductive and overwintering structures from animal predators (**Figure 9**), which guarantees a long-term survival of *A. nidulans*.

## Materials and methods

### Key resources table

| Reagent type (species) or resource | Designation | Source or reference | Identifiers | Additional information |
|---|---|---|---|---|
| Strain, strain background (*Aspergillus nidulans*) | FGSC A4 | FGSC | | Wildtype isolate (*veA+*) |

*Continued on next page*

*Continued*

| Reagent type (species) or resource | Designation | Source or reference | Identifiers | Additional information |
|---|---|---|---|---|
| Strain, strain background (*Aspergillus nidulans*) | AGB552 | *Bayram et al., 2012a* | | pabaA1; ΔnkuA::argB; veA+ |
| Strain, strain background (*Aspergillus nidulans*) | AGB596 | *Bayram and Braus, 2012b* | | $^P$gpdA:sgfp:phleo$^R$; pabaA1; yA2; veA+ |
| Strain, strain background (*Aspergillus nidulans*) | AGB1073 | This paper | | ΔlaeA::six; pabaA1; ΔnkuA::argB, veA+ |
| Strain, strain background (*Aspergillus nidulans*) | AGB1088 | This paper | | xptC:gfp; pabaA1; yA2;ΔnkuA::argB, veA+ |
| Strain, strain background (*Aspergillus nidulans*) | AGB1236 | This paper | | ΔmdpG::six; pabaA1; ΔnkuA::argB; veA+ |
| Strain, strain background (*Aspergillus nidulans*) | AGB1237 | This paper | | ΔmdpF::six; pabaA1; ΔnkuA::argB; veA+ |
| Strain, strain background (*Aspergillus nidulans*) | AGB1238 | This paper | | ΔmdpC::six; pabaA1; ΔnkuA::argB; veA+ |
| Strain, strain background (*Aspergillus nidulans*) | AGB1239 | This paper | | ΔmdpL::six; pabaA1; ΔnkuA::argB; veA+ |
| Strain, strain background (*Aspergillus nidulans*) | AGB1240 | This paper | | ΔmdpD::six; pabaA1; ΔnkuA::argB; veA+ |
| Strain, strain background (*Aspergillus nidulans*) | AGB1241 | This paper | | ΔxptA::six; pabaA1; ΔnkuA::argB; veA+ |
| Strain, strain background (*Aspergillus nidulans*) | AGB1242 | This paper | | ΔxptB::six; pabaA1; ΔnkuA::argB; veA+ |
| Strain, strain background (*Aspergillus nidulans*) | AGB1243 | This paper | | ΔxptC::six; pabaA1; ΔnkuA::argB; veA+ |
| Strain, strain background (*Aspergillus nidulans*) | AGB1248 | This paper | | ΔmdpG::six, mdpG:six::six; pabaA1; ΔnkuA::argB; veA+ |
| Strain, strain background (*Aspergillus nidulans*) | AGB1249 | This paper | | ΔmdpC::six, mdpC:six::six; pabaA1; ΔnkuA::argB; veA+ |
| Strain, strain background (*Aspergillus nidulans*) | AGB1310 | This paper | | ΔveA::six; pabaA1; ΔnkuA::argB |
| Strain, strain background (*Aspergillus nidulans*) | AGB1311 | This paper | | ΔvelB::six; pabaA1; ΔnkuA::argB, veA+ |
| Strain, strain background (*Sordaria macrospora*) | Taxid5147 | *Nowrousian et al., 2010* | | Wildtype isolate |

*Continued on next page*

*Continued*

| Reagent type (species) or resource | Designation | Source or reference | Identifiers | Additional information |
|---|---|---|---|---|
| Strain, strain background (*Verticillium longisporum*) | VL43 | *Zeise and Tiedemann, 2001* | | Wildtype isolate |
| Strain, strain background (*Verticillium dahliae*) | JR2 | *Fradin et al., 2009* | | Wildtype isolate |
| Strain, strain background (*Trichorhina tomentosa*) | | b.t.b.e. Insektenzucht GmbH | | Wildtype |
| Strain, strain background (*Folsomia candida*) | | Institute of Zoology (University of Göttingen, Germany) | | Wildtype |
| Strain, strain background (*Tenebrio molitor*) | | Zoo & Co. Zoo-Busch | | Wildtype |
| Commercial assay or kit | GeneArt Seamless Cloning and Assembly Kit | Thermo Fisher Scientific | A13288 | |
| Commercial assay or kit | GeneArt Seamless Cloning and Assembly Enzyme Mix | Thermo Fisher Scientific | A14606 | |
| Commercial assay or kit | RNeasy Plant Mini Kit | Qiagen | Cat. No./ID: 74,904 | |
| Commercial assay or kit | QuantiTect Reverse Transcription Kit | Qiagen | Cat. No./ID: 205,311 | |
| Antibody | $\alpha$-GFP antibody sc-9996 | Santa Cruz Biotechnology | RRID:AB_627695 | |
| Antibody | $\alpha$-Mouse antibody G21234 | Invitrogen AG | | |
| Software, algorithm | ImageJ | *Schneider et al., 2012* | RRID:SCR_003070 | |
| Software, algorithm | Proteome Discoverer 1.4 | Thermo Scientific | RRID:SCR_014477 | |
| Software, algorithm | Xcalibur (FreeStyle 1.4) | Thermo Scientific | RRID:SCR_014593 | |

## Strains and culture conditions

The *Escherichia coli* strain DH5α (*Woodcock et al., 1989*) was used for expression of recombinant plasmids in this study. The culture medium was Lysogeny Broth (LB) medium (*Bertani, 1951*), containing 1 % bacto tryptone, 1 % NaCl, and 0.5 % yeast extract, pH 7.5, at 37 °C. The solid LB medium was added 2 % agar additionally; 100 µg/ml of ampicillin was added as selective agent. *A. nidulans* strains were cultivated in minimal medium (MM) (AspA (70 mM NaNO$_3$, 11.2 mM KH$_2$PO$_4$, 7 mM KCl, pH 5.5), 1 % (w/v) glucose, 0.1 % (v/v) Hutner's trace elements) (*Käfer, 1977*); 2 % agar was added for the solid MM plates. For selection of *A. nidulans* transformants, phleomycin (80 µg/ml) was added. *A. nidulans* FGSC A4 was used for proteome analysis. *A. nidulans* AGB552 (*Bayram et al., 2012a*) was used for the construction of *A. nidulans* mutants. Cultivation of strains based on AGB552 required the addition of 0.0001 % 4-aminobenzoic acid.

For vegetative growth, *A. nidulans* spores were inoculated and shaken 20 hr in liquid MM at 37 °C in flasks with baffle. For asexual development, *A. nidulans* spores were incubated on solid MM plates at 37 °C under illumination. For sexual development, the solid MM plates were sealed with Parafilm M (Merck, Darmstadt, Germany,) and incubated at 37 °C in dark. Conidia were stored in 0.96 % NaCl buffer with 0.02 % Tween-80 (Sigma-Aldrich Chemie GmbH, Taufkirchen, Germany) at 4 °C.

*S. macrospora* was cultivated on cornmeal malt fructification medium (BMM) agar plates at 27 °C (*Dirschnabel et al., 2014*). *Verticillium* spp. were cultivated on simulated xylem medium (SXM) agar plates at 25 °C (*Hollensteiner et al., 2016*). All fungal strains are listed in *Supplementary file 3*.

## Plasmid and *A. nidulans* strain construction

The GeneArt Seamless Cloning and Assembly Enzyme Mix (Thermo Fisher Scientific, Waltham, MA) or the GeneArt Seamless Cloning and Assembly Kit (Thermo Fisher Scientific) was used for the assembly of DNA fragments with the backbone. For DNA amplification, gDNA from *A. nidulans* AGB552 strain was used. The constructed and used plasmids in this study are present in *Supplementary file 4*. The used primers are present in *Supplementary file 5*.

The recyclable marker cassette-containing plasmid pME4319 was used as the cloning vector, which harbors the *bleo* gene conferring resistance to phleomycin (*Drocourt et al., 1990*). For pME4319 construction, the pBluescript II (+) vector was amplified with primers flip1, carrying a *Swa*I site, and flip2, carrying a *Pml*I site. The recyclable phleomycin resistance marker was cut from pME4305 with *Sfi*I (*Thieme et al., 2018*). Both fragments were assembled as described above. For the *Aspergillus* deletion strain generation, approximately 1.5 kb of the 5′ flanking region of the gene of interest, carrying a *Pme*I site, was inserted into the *Swa*I site of pME4319 and approximately 1.5 kb of the 3′ flanking region of the gene of interest, carrying a *Pme*I site, was inserted into the *Pml*I site of pME4319. For 'on-locus' transformation, the cassette containing the 5′ flanking region, the recyclable phleomycin resistance marker, and the 3′ flanking region was cut as a linear fragment by *Pme*I.

The 5′ flanking region (1.5 kb, primers LL139/140) and 3′ flanking region of *mdpG* (1.6 kb, primers LL141/142) were inserted into the *Swa*I and *Pml*I sites of pME4319, respectively, resulting in pME4842. The cassette of Δ*mdpG* was excised and integrated into AGB552 for the *mdpG* deletion strain AGB1236. The 5′ flanking region (1.0 kb, primers LL197/198) and 3′ flanking region of *mdpF* (1.0 kb, primers LL199/200) were integrated into pME4319, resulting in pME4843. The cassette of Δ*mdpF* was integrated into AGB552 for the *mdpF* deletion strain AGB1237. The 5′ flanking region (1.0 kb, primers LL223/224) and 3′ flanking region of *mdpC* (1.1 kb, primers LL225/226) were integrated into pME4319, resulting in pME4844. The deletion cassette was integrated into AGB552 for the *mdpC* deletion strain AGB1238. The 5′ flanking region (1.2 kb, primers LL193/194) and 3′ flanking region of *mdpL* (1.2 kb, primers LL195/196) were inserted into pME4319, giving rise to pME4845. The Δ*mdpL* cassette was integrated into AGB552 for the *mdpL* deletion strain AGB1239. The 5′ flanking region (1.1 kb, primers LL162/163) and 3′ flanking region of *mdpD* (1.3 kb, primers LL164/165) were cloned into pME4319, giving rise to pME4846. The deletion cassette was integrated into AGB552 for the *mdpD* deletion strain AGB1240. The 5′ flanking region (1.3 kb, primers LL211/212) and 3′ flanking region of *xptA* (1.3 kb, primers LL213/214) were cloned into pME4319, resulting in pME4847. The Δ*xptA* cassette was integrated into AGB552 for the *xptA* deletion strain AGB1241. The 5′ flanking region (1.5 kb, primers LL215/216) and 3′ flanking region of *xptB* (1.3 kb, primers LL217/218) were cloned into pME4319, resulting in pME4848. The Δ*xptB* cassette was excised and integrated into AGB552 for the *xptB* deletion strain AGB1242. The 5′ flanking region (1.0 kb, primers LL219/220) and 3′ flanking region of *xptC* (1.1 kb, primers LL221/222) were integrated into pME4319, giving rise to pME4849. The cassette of Δ*xptC* was excised and integrated into AGB552 for the *xptC* deletion strain AGB1243.

For the construction of the *mdpG* complementation strain, 1.5 kb of the 5′ flanking region as well as the *mdpG* gene were amplified in two PCRs with primers LL59/60 and primers LL61/62. Both fragments were inserted into the *Swa*I site of pME4319; 1.6 kb of 3′ flanking region of *mdpG* (primers JG1076/EFS46) was inserted into the *Pml*I site, resulting in pME4850. The *mdpG* complementation cassette was excised from pME4850 and integrated into AGB1236 for AGB1248. For construction of *mdpC* complementation strain, 1.9 kb of 5′ flanking region and the *mdpC* ORF (primers LL223/236) was inserted into the *Swa*I site of pME4319; 1.1 kb of 3′ flanking region of *mdpC* (primers LL225/226) was inserted into the *Pml*I site, giving rise to pME4851. The complementation cassette of *mdpC* was integrated into AGB1238 for AGB1249.

For the construction of *xptC:gfp*, 3.5 kb of 5′ flanking region and the *xptC* ORF was amplified with the primers BD113/114 and fused with *gfp* (primers BD106/107). The fragment was inserted into the *Swa*I site of pME4319; 3.8 kb of 3′ flanking region amplified with the primers BD115/116 was inserted into the *Pml*I site, giving rise to pME4645. The *xptC:gfp* cassette was introduced into AGB552, resulting in AGB1088.

For the construction of AGB1310, the *veA* deletion cassette was excised from pME4574 with *Pme*I and integrated into AGB552. For the construction of AGB1311, the *velB* deletion cassette was excised from pME4605 with *Pme*I and integrated into AGB552.

For the construction of pME4636, 2.3 kb of *laeA* 5' flanking region was amplified with primers BD45/BD46 and 2.1 kb of *laeA* 3' flanking region was amplified with primers BD47/BD48. The fragments were inserted into pME4319. The *laeA* deletion cassette was excised with *Pme*I from pME4636 and integrated into AGB552, resulting in AGB1073.

### Transformation of *E. coli* and *A. nidulans*

*E. coli* transformation was performed using the heat-shock method (*Inoue et al., 1990*). *A. nidulans* transformation was performed using the polyethylene glycol-mediated protoplast fusion method (*Punt and van den Hondel, 1992*). Successful transformation was further verified by *Southern, 1975*, hybridization. The recyclable marker cassette was eliminated from the genome by culturing the fungus on 1 % xylose MM plate (*Hartmann et al., 2010*).

### Semi-quantitative reverse-transcriptase polymerase chain reaction

Fungal tissues were harvested from cultures, which were incubated under conditions inducing the sexual cycle, after 3 days of incubation and ground with a table mill in powder. The RNA isolation was performed according to the instruction of the RNeasy Plant Miniprep Kit (Qiagen, Hilden, Germany). Approximately 0.8 µg of RNA was used for cDNA synthesis according to manufacturer's instructions of the QuantiTect Reverse Transcription Kit (Qiagen); 1 µl of cDNA was used for semi-quantitative reverse-transcriptase polymerase chain reaction. The used primers are listed in the *Supplementary file 6*. Measurements were conducted in three independent biological replicates.

### Protein extraction of *A. nidulans*

Asexual and sexual mycelia of *A. nidulans* wildtype (FGSC A4, *veA*+) were harvested from solid agar plate cultures after 3, 5, and 7 days. Vegetative mycelia were harvested from 20 hr of liquid culture. Mycelia samples were frozen in liquid nitrogen and ground with a table mill. Protein extraction was carried out as described (*Bayram et al., 2012c*). Hülle cells were collected from 3-, 5-, and 7-day-old cleistothecia, grown under conditions inducing the sexual cycle, by rolling cleistothecia on an agar plate surface. Hülle cells were sonicated for cell disruption repeatedly (60 % of sonication power for 60 s, in between centrifugation for 30 s at 4 °C) and centrifuged for 10 min (13,000 rpm, 4 °C). The supernatant-containing proteins was harvested for further experiments. For each sample, three biological replicates were prepared.

### Protein digestion with trypsin and LC-MS analysis

Approximately 80 µg of protein were separated by SDS-PAGE for 60 min at 200 V. The gel was stained (*Neuhoff et al., 1988*) and the lanes were excised and subjected to tryptic digestion (*Shevchenko et al., 1996*). Digested peptides were desalted by using C18 stage tips (*Rappsilber et al., 2007*). The peptides were resuspended in 20 µl sample buffer (2 % [v/v] acetonitrile and 0.1 % [v/v] formic acid). LC-MS was performed by using a Velos Pro Hybrid Ion Trap-Orbitrap mass spectrometer (MS) coupled to a Dionex Ultimate 3000 HPLC (Thermo Fisher Scientific) (*Schmitt et al., 2017*). Proteomics raw data were searched with SEQUEST and Mascot algorithms present in Proteome Discoverer 1.4 using the *A. nidulans* genome database (AspGD) (*Cerqueira et al., 2014*; *Eng et al., 1994*; *Koenig et al., 2008*). The search parameter for the algorithms were: 10 ppm of precursor ion mass tolerance; 0.6 Da of fragment ion mass tolerance; two maximum of missed cleavage sites; variable modification by methionine oxidation; fixed cysteine static modification by carboxyamidomethylation. Results filter settings: high peptide confidence; minimal number of two peptides per protein.

### Fluorescence microscopy of fusion proteins

Vegetative mycelia (20 hr) as well as 3-day-old sexual mycelia and Hülle cells were transferred to an object slide. Fluorescence microphotographs were taken with a confocal light microscope (Zeiss Axiolab-Zeiss AG) equipped with a QUAN-TEM: S12SC (Photometrics) digital camera and the software package SlideBook 6 (Intelligent Imaging Innovations GmbH, Göttingen, Germany).

### Immunoblotting

Extracted proteins of 3-day-old sexual fungal tissues were separated by PAGE and transferred onto a nitrocellulose membrane (GE Healthcare, Wauwatosa, WI) as described (*Schinke et al., 2016*).

Ponceau staining was used as sample loading control. After blocking in 5 % skim milk powder dissolved in TBS-T buffer, the first antibody α-GFP antibody (sc-9996, Santa Cruz Biotechnology Inc, Santa Cruz, CA) was diluted 1:1000 in blocking solution and the second α-mouse antibody (G21234, Invitrogen AG) was diluted 1:2000 in blocking solution. The result was visualized on a Fusion-SL7 (Vilber Lourmat, Collégien, France) system. The experiments were carried out with three biological replicates.

## Extraction of SMs and LC-MS analysis

Four microliter containing approximately 1000 conidia were point-inoculated on MM agar plates and sexually grown for 2, 3, 5, 7, and 10 days. A 5.7 cm$^2$ agar piece of the colonies was cut into small pieces and covered with 5 ml of ethyl acetate in 50 ml tube. Tubes were shaken at 200 rpm at room temperature for 30 min followed by 10 min highest level ultra-sonication in a Bandelin Sonorex Digital 10 P ultrasonic bath (Bandelin Electronic GmbH & Co. KG, Berlin, Germany); 3 ml of ethyl acetate phase was transferred to a glass tube and evaporated. In a similar procedure, the secreted metabolites were extracted, with small modifications. The fungal colony of each strain was removed using a scalpel. Afterward, a 5.7 cm$^2$ piece of agar of the region where the colony was grown was cut and used for extraction as mentioned above. For this experiment, mycelium was grown under sexual-inducing conditions for 3 days.

SM sample was suspended in 700 µl methanol (500 µl for the extracts received from the secretion experiments) and centrifuged for 10 min (13,000 g, 4 °C) to remove particles; 500 µl (400 µl for secretion extracts) of supernatant was transferred into the LC-MS vial. LC-MS was performed by using a Q Exactive Focus orbitrap mass spectrometer coupled to a Dionex Ultimate 3000 HPLC (Thermo Fisher Scientific); 5 µl of SM sample was injected into the HPLC column (Acclaim 120, C$^{18}$, 5 µm, 120 Å, 4.6 × 100 mm). The running phase was set as a linear gradient from 5% to 95% (v/v) acetonitrile/0.1 formic acid in 20 min, plus 10 min with 95 % (v/v) acetonitrile/0.1 formic acid with a flow rate of 0.8 ml/min at 30 °C in addition. The measurements were performed in positive and negative modes with a mass range of m/z 70–1050 . FreeStyle 1.4 (Thermo Fisher Scientific) was used for data analysis.

## Monitoring of sexual development

Approximately 1000 conidia of A. nidulans strains were point-inoculated on MM agar plates and cultivated under sexual conditions. Sexual fruiting body development of each strain was monitored at 2, 3, 4, 5, 7, and 10 days. The development status of the cleistothecia at the colony center was recorded over time with photomicrographs. Matured cleistothecia were collected and counted, and their diameters were measured by a microscope with the software cellSens Dimension (Olympus Europa SE & Co. KG, Hamburg, Germany). Cleistothecia were broken in 100 µl 0.02 % Tween buffer for ascospore quantification (n = 10). This was performed with three biological and three technical replicates. Hülle cells were detached from 5-day-old cleistothecia by rolling the fruiting bodies on agar surface. The diameter of Hülle cells was measured as described for the cleistothecia. Two independent experiments were carried out.

## Analysis of germination of Hülle cells

Hülle cells were collected from 5-day-old cleistothecia of AGB552, ΔmdpG, ΔmdpC, and ΔmdpL separately. Hülle cells were picked with an MSM System 300 micromanipulator (Singer Instruments) and placed on separate fresh MM plates (n = 40 (±1)). For germination experiments, plates were incubated for 2 days under light conditions at 37 °C. The germination rate of detached Hülle cells was calculated from the visible colonies after 48 hr. This was performed in two biological replicates.

## Effect of secondary metabolite on fungi

SMs of mdp/xpt gene deletion strains and A. nidulans wildtype AGB552, extracted from nine point-inoculated colonies, were dissolved in 450 µl of methanol; 30 µl of mixed solution was loaded on the filter paper disc (Φ = 9 mm) individually, 30 µl of pure methanol was used as a blank control, 75 µg of pure ω-hydroxyemodin (ChemFaces, Wuhan, China), emodin (VWR, Darmstadt, Germany) and chrysophanol (VWR) were dissolved in methanol and loaded on the paper disc for following tests. 75 µg emodin roughly correlates with the amount received from 6 to 7 A. nidulans ΔmdpC colonies after 3 days of growth.

Loaded paper discs were placed on plates inoculated with spores of the tested fungi. For *A. nidulans* wildtype AGB552, $1 \times 10^5$ fresh conidia were spread on 80 ml MM agar plates and incubated under sexual growth-inducing conditions for 5 days. The size and amount of cleistothecia on each paper disc were monitored. For *S. macrospora*, $2 \times 10^5$ spores were spread on cornmeal malt fructification medium (BMM) agar plates and grown for 7 days at 27 °C. For *Verticillium* spp., $1 \times 10^5$ spores were separated completely on SXM agar plates and grown for 10 days at 25 °C. Race tube experiments were carried out as follows: 20 ml BMM agar were supplemented with 2 mg total extracts of the wildtype AGB552 and the Δ*mdpC* strain (diluted in 140 µl methanol). As control 140 µl methanol was used. Small agar pieces of *S. macrospora* wildtype was put in the race tubes and pre-incubated for 3 days. Starting on the third day, the growth rate was measured in the race tubes for 7 days. The complete experiment was carried out in the presence of light at 27 °C. Three biological replicates with at least two technical replicates were used. Three biological replicates were used for the methanol control.

## Animal food preference

*T. molitor* larvae were purchased from Zoo & Co. Zoo-Busch (Göttingen, Germany). *F. candida* was provided by the Institute of Zoology (University of Göttingen, Germany), and was kept on the plaster Petri dishes (gypsum plaster: charcoal [9:1]) (*Xu et al., 2019*). *T. tomentosa* was purchased from b.t.b.e. Insektenzucht GmbH (Schnürpflingen, Germany). *A. nidulans* spores were point-inoculated on 50 ml MM agar plates and incubated for 5 days of sexual growth.

For the experiment with *T. molitor*, fungal colony agar pieces (Φ = 2.7 cm) were cut and placed on two opposite sides of a Petri dish (140 mm in diameter). Animals (n = 10) were placed into the center area of the Petri dish. The number of animals on each side was counted over a period of 8 hr. The experiment was carried out with three biological and nine technical replicates.

For the experiment with *F. candida*, the *A. nidulans* colony agar pieces (Φ = 1.5 cm) were placed onto opposite sides of the plaster Petri dishes (92 mm in diameter). Approximately 20 animals were placed onto the center of the Petri dish and the number on each side was counted over a period of 24 hr. The experiment was performed with three biological and four technical replicates.

The food choice experiment with the isopod *T. tomentosa* was carried out as described for *F. candida* with n = 8 animals.

To analyze the feeding behavior of the animal predators, $1 \times 10^6$ spores of the *A. nidulans* strains were plated on MM and incubated under sexual-inducing conditions for 5 days. Agar pieces (Φ = 2.7 cm) were stamped out with a Falcon tube and put in a Petri dish. Approximately 30 animals of *F. candida* were incubated with the agar piece for 6 days and scanned. A quantification of these plates was not possible because, the eaten parts were often too diffuse for analyzing. However, differences in the feeding behavior of the animals on the different mycelia could be observed by eye. The same experiments were carried out with *T. tomentosa* with n = 5 animals per agar piece. The software ImageJ (*Schneider et al., 2012*) was used to define the mycelium-free parts of the different agar pieces for the experiments with *T. tomentosa*. Only mycelium-free parts with more than 1 % of the total agar piece area were taken for the analysis. Two independent experiments were performed. Each experiment consisted of at least six different agar pieces from at least two independent plates. In total 16 agar pieces were analyzed for *T. tomentosa* and 14 for *F. candida*.

Cleistothecia experiments were carried out as follows: Cleistothecia were isolated with Hülle cells from mycelium induced under sexual conditions for 4 days and were put on a water/agar plates without any additional compounds. Each water/agar plate contained 10 cleistothecia. In total, six agar plates (n = 60 cleistothecia) were used for each experiment. Two independent experiments were carried out with *T. tomentosa* (five animals per plate) and four independent experiments with *F. candida* (approximately 20 animals per plate). After 24 hr the remaining cleistothecia were counted.

## Toxicity test

SMs were extracted from plates which were incubated for 4 days under conditions inducing the sexual cycle. Received extracts were solved in methanol; 100 µg SMs per mg animal food were used. The amounts of metabolites were determined using an LC-MS Q Exactive Focus orbitrap mass spectrometer coupled to a Dionex Ultimate 3000 HPLC (Thermo Fisher Scientific). CAD was used as detector. As standard, para-aminobenzoic acid was added to the samples in a final concentration of 0.002 %.

For quantification, the FreeStyle software version 1.6 (Thermo Fisher Scientific) was used. The wild-type extract contains approximately 75 µg/mg emericellin/shamixanthone and 32.5 µg/mg epishamixanthone. Animal food was dried for 3 days to evaporate the methanol. As control, food was mixed with methanol without SMs and dried as well. Animals were fed with the prepared food for 5 days; 100 µg total extracts per mg food was used for the experiments. Experiments were carried out on 24-well plates at 22 °C. Before feeding the animals were starved for 20 hr. Petri dishes were sealed with parafilm. For *T. tomentosa* oat flakes and for *F. candida* dried yeast were used as food. For *T. tomentosa* four animals were used per strain. For *F. candida* six different experiments were carried out except for Δ*mdpG* with five independent experiments. Each experiment contains 1–12 animals. Total number of animals were n = 22 for the control, n = 26 for Δ*mdpG*, n = 44 for wildtype, and n = 43 for Δ*mdpC*. The survival rate was determined after 5 days. Significance was calculated using the two-tailed t-test with $p < 0.05$.

## Acknowledgements

We thank Verena Große, Nicole Scheiter, Gertrud Stahlhut, and Ruth Pilot for technical assistance, Kerstin Schmitt and Miriam Kolog Gulko for preparing protein digestion solutions and proteome data discussions. We acknowledge support by the doctoral programs of Göttinger Graduiertenzentrum für Neurowissenschaften, Biophysik und Molekulare Biowissenschaften (GGNB) (University of Göttingen), the China Scholarship Council (CSC), and the European Union's Seventh Framework Programme FP7/2007-2013 (grant agreement 607332). Funding was provided by the German Research Council to GHB (DFG grant BR 1502/19–1).

## Additional information

### Funding

| Funder | Grant reference number | Author |
|---|---|---|
| Deutsche Forschungsgemeinschaft | BR 1502/19-1 | Gerhard H Braus |
| Seventh Framework Programme | 607332 | Gerhard H Braus |
| Göttingen Graduate School for Neurosciences, Biophysics, and Molecular Biosciences | | Li Liu |
| China Scholarship Council | | Li Liu |

The funders had no role in study design, data collection and interpretation, or the decision to submit the work for publication.

### Author contributions

Li Liu, Conceptualization, Data curation, Formal analysis, Investigation, Methodology, Software, Validation, Visualization, Writing – original draft; Christoph Sasse, Benedict Dirnberger, Data curation, Formal analysis, Investigation, Software, Visualization, Writing – original draft; Oliver Valerius, Data curation, Formal analysis, Investigation, Methodology, Software, Writing – original draft; Enikő Fekete-Szücs, Investigation, Writing – original draft; Rebekka Harting, Formal analysis, Investigation, Methodology, Writing – original draft; Daniela E Nordzieke, Formal analysis, Investigation, Writing – original draft; Stefanie Pöggeler, Conceptualization, Formal analysis, Methodology, Resources, Supervision, Writing – original draft; Petr Karlovsky, Conceptualization, Data curation, Formal analysis, Methodology, Resources, Supervision, Writing – original draft; Jennifer Gerke, Conceptualization, Data curation, Formal analysis, Investigation, Methodology, Project administration, Software, Supervision, Validation, Visualization, Writing – original draft; Gerhard H Braus, Conceptualization, Funding acquisition, Project administration, Resources, Supervision, Writing – original draft

## Author ORCIDs

Christoph Sasse http://orcid.org/0000-0003-2612-1313
Benedict Dirnberger http://orcid.org/0000-0002-0772-5923
Daniela E Nordzieke http://orcid.org/0000-0001-6621-2565
Stefanie Pöggeler http://orcid.org/0000-0002-6842-4489
Petr Karlovsky http://orcid.org/0000-0002-6532-5856
Jennifer Gerke http://orcid.org/0000-0001-7301-2676
Gerhard H Braus http://orcid.org/0000-0002-3117-5626

## Decision letter and Author response

Decision letter https://doi.org/10.7554/eLife.68058.sa1
Author response https://doi.org/10.7554/eLife.68058.sa2

## Additional files

### Supplementary files

• Supplementary file 1. LC-MS analysis revealed 24 proteins that were exclusively produced in both sexual mycelia and Hülle cells.

• Supplementary file 2. Secondary metabolites produced by the *mdp/xpt* genes in *Aspergillus nidulans* sexual development identified by LC-MS.

• Supplementary file 3. Fungal strains used in this study.

• Supplementary file 4. Plasmids employed in this study.

• Supplementary file 5. Primers for DNA sequence amplification and plasmid construction.

• Supplementary file 6. Primers for semi-quantification of the *mdp/xpt* cluster genes.

• Transparent reporting form

### Data availability

All data generated during this study are included in the manuscript and supporting files. Source data files have been provided.

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
