## [Decision Letter]

**Acceptance summary:**

Hülle cells, a type of cells formed by fungal species of the genus Aspergillus, are specialized cells that surround the sexual fruiting bodies and nurse them during fungal sexual development. In this work, Liu et al., suggest that these cells have a strong ecological impact because they contain specific secondary metabolites that help the fungus to "withstand" the attack by fungivorous animals, like springtails, and also inhibit sexual reproduction of other fungi. This work sheds new light on the development and ecology of fungi as well as on the ecological functions of secondary metabolites.

**Decision letter after peer review:**

Thank you for submitting your article "Hülle-cell-mediated protection of fungal reproductive and overwintering structures against fungivorous animals" for consideration by *eLife*. Your article has been reviewed by 3 peer reviewers, and the evaluation has been overseen by a Reviewing Editor and Detlef Weigel as the Senior Editor. The following individuals involved in review of your submission have agreed to reveal their identity: Milton Drott (Reviewer #2); Jae-Hyuk Yu (Reviewer #3).

Essential revisions:

There was enthusiasm by all three reviewers for this work and several constructive suggestions were raised by each of the reviewers. In your revisions, there are two items that must be addressed:

1) The insect-fungus interactions will require additional experiment(s) to clarify if differences in "food preference" actually are associated with difference in fungivory. Given central hypotheses that the secondary metabolite compounds are cleistothecia/hulle-cell associated, the reviewers would like to see how the compounds of interest may impact the fitness of cleistothecia specifically. At present, it is not clear if differences in insect behavior associated with these compounds have any impact on the fungus.

2) There is a need for clarification of some of the ecological inferences on the fungus-fungus interaction. These can be dealt by being more explicit about the limitations of the experiments and inferences drawn and by clarifying how the concentrations of compounds used relate to what might actually occur in/around (secreted and sequestered in structures) the fungus.

*Reviewer #1 (Recommendations for the authors):*

I have a few specific comments:

1. The effect of epi-/shamixanthone precursors on fruiting body formation and resting structure formation of other fungi. An alternative hypothesis could be that the compounds just inhibit the growth of these fungi. Although the finding is very compelling, I missed a control experiment like measurement of hyphal extension of these fungi in absence / presence of crude extracts and purified compounds.

2. Figure 1: In my version, the pictures and magnifications were too small; in particular xptC:gfp, fluorescence; this figure would need some improvement.

3. Table 1: Could you show the number of spectra found in vegetative hyphae. If it were zero, just mention it.

4. References often include the first name or abbreviation of the first name, please delete.

5. Any mechanistic hypothesis why the size of Hülle cells is decreased due to accumulation of SM precursors? Maybe they are just toxic?

6. What is the hypothesis that the intact *mdp*/*xpt* cluster is required for accurate sexual development?

7. How to explain delayed germination, when the compounds are missing? Or the accumulated intermediates are toxic?

8. Figure 9 is a nice summary of the findings, but can be improved by better arrangements.

*Reviewer #2 (Recommendations for the authors):*

The introduction contains much of the relevant information needed to approach the paper but does not flow clearly from secondary metabolism to sexual reproduction (and the subsequent importance of Hulle cells). It is my understanding that the authors try to connect these ideas through the velvet complex. While the velvet complex does indeed impact sexual reproduction and secondary metabolism, LaeA deletion mutants are often found to differentially express thousands of genes. Thus, associations between Hulle cells and SM formation that are based solely on this global regulator do not imply any close relationship. I would recommend decreasing the focus on the velvet complex.

Related to the previous point, while velvet-complex mutants have been of great value for disentangling molecular pathways and understanding the general architecture of gene regulation in fungi, they are of extremely limited value to ecological experiments. I appreciate that for consistency with other papers this may be a reasonable piece of data to retain. However, it is important that the authors acknowledge that there are severe limits on what inferences we can draw from these mutants. As the authors note (line 473), the velvet complex is conserved across fungi, indicating a strong selective pressure for this complex. Mutants of this complex do not grow normally. Thus, ecological inferences based on these mutants are based on genotypes that would not persist in nature and are thus of dubious value. Furthermore, given that these mutants are often aberrant in the expression of thousands of genes, interpretation of any resulting phenotypes is near impossible. While other studies have used similar approaches, I have always been extremely skeptical of ecological inferences that stem from velvet-complex mutants.

The authors do support their claims with specific mutants of the cluster that they are looking at. I applaud them for also using appropriate complementation controls.

The authors do a very nice job of showing the localization of mdp/xpt cluster metabolites and their gene expression. I would recommend adding the non-sexual mycelial images that are currently in the supplement to figure 1. Similarly, Table 1 should also include values for non-sexual mycelia.

The use of the term "accurate" for sexual development is a little misleading as it implies that something is inherently wrong (i.e., "inaccurate"). It appears that while the number and size of cleistothecia are impacted by intermediates in this pathway, it does seem that this structure, and the ascospores are produced. I agree that these changes may be important, but I think the language surrounding this discussion needs to be more nuanced to better reflect the possible ecological impact (or lack thereof) of these differences.

Even when appropriate complementation controls are used, there can be concerns about the use of mutants for ecological inferences that touch on fitness. In particular, some associations between secondary metabolism and fungal fitness that have been elucidated with mutants seem to have no reflection on natural patterns. For example, using mutants it has been shown that conidiation is decreased when aflatoxin production is knocked out. However, studies have not been able to find any evidence that non-aflatoxigenic isolates of Aspergillus spp. produce fewer conidia (even when comparing closely related isolates). The transformation process alone (without disrupting genes) can result in shifts in fitness (https://doi.org/10.1094/Phyto-80-1166) – not to mention off-target impacts of gene deletion. It is thus of great importance to consider how the mutations induced in this study might reflect mutations that occur and or persist in nature. Problematically, the WT does not accumulate many of the compound intermediates (figure S10) that are determined to inhibit the growth of other fungi, raising questions about the relevance of these experiments to nature. While it is fine to look at the impact of these intermediates, the authors must acknowledge the fact that these intermediates may not occur in nature and that isolates that do produce them may succumb to selective pressures that could be associated with their findings of aberrant sexual development (some clarification on how the authors can be sure that the aberrant cleistothetical development is bad would benefit the manuscript – I don't know that I disagree, but it is worth discussing).

Interpretation of the fungal-competition experiments is further complicated by questions that remain about the concentrations of pure/crude compounds used. Were concentrations of extracts at all reflective of what might be produced/secreted by A. nidulans? Indeed – are these compounds secreted at all? How does the 75 ug per disc reflect concentrations actually produced by the fungus? Can anything resembling inhibitory concentrations be produced on non-synthetic media that may be encountered in nature? Importantly if these compounds are not secreted, might they instead protect against the growth of other fungi directly on cleistothecia?

I commend the authors for their use of three diverse fungivores. However, the simple food-choice test that was conducted does not provide much insight into the ecological role of xanthones. It is important to note that just because insects spend more time near colonies unable to produce *mdp*/*xpt* metabolites does not necessarily mean that these colonies are eaten more or have a fitness disadvantage. Other studies approaching similar questions, including those cited by the authors, have performed far more comprehensive assays to draw similar conclusions as the authors do here. For example, Cary et al. (2014) performed similar food-choice tests of fungal tissues to fungivores. However, in addition to offering fungivores a choice between two genotypes of fungus, they also performed no-choice experiments that help to elucidate the nature of antifeedant activity. Furthermore, Cary et al. assessed the weight of fungal structures in question, thus adding a fitness measure to their food-choice. It is also not uncommon for studies to also assess the toxicity of compounds of interest on insects.

The authors approach to the insect experiments is a little confusing in light of their hypothesis. It is my understanding of the authors argument that xanthones are targeted through Hülle cells to protect cleistothecia. However, the authors do not investigate if there is any difference in the protection conferred specifically to these structures. If xanthones do indeed have strong antifeedant effects, then it would be important to clarify if these effects are conferred preferentially to the cleistothecia or if they are realized by the entire thallus.

The authors identify a correlation between species that produce Hülle cells and those that produce the *mdp*/*xpt* metabolites. While this is intriguing, it is not evidence that these other species also produce mdp/xpt metabolites in a Hülle-cell associated manner. The text should be more cautious in its interpretation (lines 479-485).

The discussion should acknowledge a large body of literature that now ties close connections between secondary metabolism and primary growth raising questions about the authors suggestion that SMs are not "directly involved in growth".

I do not find the speculation about the Emodin target very convincing. Additionally, I do not find conclusions about the ecological role of emodin production convincing given that emodin does not seem to be accumulated by the WT and thus the ecological function of emodin production rests in its derivatives.

The manuscript is clear, well written, and well organized. There are a few minor grammatical errors throughout that could be fixed to make the manuscript feel even more polished. I acknowledge that many of these errors are subtle and may not be immediately apparent even for native English speakers. For example (but not limited to): (a) line 124 "specifically located to" should be "localized to" (b) line 206 "this assisted us to trace" should be "this enabled us to trace" (c) line 304-305 "repression of cleistothecia size" should be "decrease in cleistothecial size".

---

## [Author Response]

Essential revisions:There was enthusiasm by all three reviewers for this work and several constructive suggestions were raised by each of the reviewers. In your revisions, there are two items that must be addressed:1) The insect-fungus interactions will require additional experiment(s) to clarify if differences in "food preference" actually are associated with difference in fungivory. Given central hypotheses that the secondary metabolite compounds are cleistothecia/hulle-cell associated, the reviewers would like to see how the compounds of interest may impact the fitness of cleistothecia specifically. At present, it is not clear if differences in insect behavior associated with these compounds have any impact on the fungus.

We addressed this issue by carrying out additional experiments, which revealed that secondary metabolites of *A. nidulans* are not toxic to the fungivorous springtail *Folsomia candida*. We were able to show that metabolites of the *A. nidulans* wildtype protected the cleistothecium as well as sexual mycelium. These results, including feeding behavior on mycelium of different strains as well as images from the toxicity tests are presented in the new Figure 8.

2) There is a need for clarification of some of the ecological inferences on the fungus-fungus interaction. These can be dealt by being more explicit about the limitations of the experiments and inferences drawn and by clarifying how the concentrations of compounds used relate to what might actually occur in/around (secreted and sequestered in structures) the fungus.

We addressed fungus-fungus interactions by additional experiments, which demonstrated that the *A. nidulans* secondary metabolites do only affect the fruiting bodies, but not the vegetative growth of the fungus *S. macrospora*. These data were included as Figure 6-supplement figure 1b. We used 75 µg of pure compounds per disc, which corresponds to the amount of emodin, which was on average extracted from 6-7 point-inoculated *A. nidulans* deletion mutant colonies with a size of 5.7 cm^2^ after 3 days of growth. We also added experiments which revealed that the identified metabolites can be secreted to the medium and included these data as Figure 2—figure supplement 4. It remains to be shown during what conditions *A. nidulans* might secrete these compounds in its natural habitats to protect cleistothecia and sexual mycelia.

Reviewer #1 (Recommendations for the authors):I have a few specific comments:1. The effect of epi-/shamixanthone precursors on fruiting body formation and resting structure formation of other fungi. An alternative hypothesis could be that the compounds just inhibit the growth of these fungi. Although the finding is very compelling, I missed a control experiment like measurement of hyphal extension of these fungi in absence / presence of crude extracts and purified compounds.

We performed the suggested control experiment for the fungal interaction between *A. nidulans* and *S. macrospora* vegetative hyphae*.* Metabolites of extracts of the *ΔmdpC A. nidulans* strain were applied, which inhibit Sordaria fruiting body formation. These extracts specifically affected development but not the vegetative growth rate of *S. macrospora* (lines 399-401, Figure 6—figure supplement 1b and Figure 6—figure supplement 1-source data 2, lines 25-26 in abstract). The experimental set up was added to the description of methods (lines 827-834).

2. Figure 1: In my version, the pictures and magnifications were too small; in particular xptC:gfp, fluorescence; this figure would need some improvement.

The figure was enlarged as suggested for better visualization of the cellular XptC-GFP localization (Figure 1).

3. Table 1: Could you show the number of spectra found in vegetative hyphae. If it were zero, just mention it.

We added the spectra as suggested to Table 1 and the table legend (lines 130-132).

4. References often include the first name or abbreviation of the first name, please delete.

References were corrected as suggested.

5. Any mechanistic hypothesis why the size of Hülle cells is decreased due to accumulation of SM precursors? Maybe they are just toxic?

The SM precursors might inhibit Hülle cell growth and germination. We included these points into the manuscript (lines 238-239).

6. What is the hypothesis that the intact *mdp*/*xpt* cluster is required for accurate sexual development?

The *mdp*/*xpt* cluster metabolites change over time during fruiting body development and are also secreted into the medium (lines 198-202). Addition of precursors of the *mdp*/*xpt* cluster inhibits sexual development (lines 317-321). One attractive hypothesis is that the intact *mdp*/*xpt* cluster is required to reduce precursors, which inhibit sexual development (added on lines 323-325).

7. How to explain delayed germination, when the compounds are missing? Or the accumulated intermediates are toxic?

The SM precursors might inhibit or even be toxic for Hülle cell germination. We included this point into the manuscript (lines 238-239)

8. Figure 9 is a nice summary of the findings, but can be improved by better arrangements.

We rearranged Figure 9 as suggested.

Reviewer #2 (Recommendations for the authors):The introduction contains much of the relevant information needed to approach the paper but does not flow clearly from secondary metabolism to sexual reproduction (and the subsequent importance of Hulle cells). It is my understanding that the authors try to connect these ideas through the velvet complex. While the velvet complex does indeed impact sexual reproduction and secondary metabolism, LaeA deletion mutants are often found to differentially express thousands of genes. Thus, associations between Hulle cells and SM formation that are based solely on this global regulator do not imply any close relationship. I would recommend decreasing the focus on the velvet complex.

The paragraph was modified as suggested (lines 75-79).

Related to the previous point, while velvet-complex mutants have been of great value for disentangling molecular pathways and understanding the general architecture of gene regulation in fungi, they are of extremely limited value to ecological experiments. I appreciate that for consistency with other papers this may be a reasonable piece of data to retain. However, it is important that the authors acknowledge that there are severe limits on what inferences we can draw from these mutants. As the authors note (line 473), the velvet complex is conserved across fungi, indicating a strong selective pressure for this complex. Mutants of this complex do not grow normally. Thus, ecological inferences based on these mutants are based on genotypes that would not persist in nature and are thus of dubious value. Furthermore, given that these mutants are often aberrant in the expression of thousands of genes, interpretation of any resulting phenotypes is near impossible. While other studies have used similar approaches, I have always been extremely skeptical of ecological inferences that stem from velvet-complex mutants.

We addressed this important point into discussion (lines 565-575).

The authors do support their claims with specific mutants of the cluster that they are looking at. I applaud them for also using appropriate complementation controls.The authors do a very nice job of showing the localization of mdp/xpt cluster metabolites and their gene expression. I would recommend adding the non-sexual mycelial images that are currently in the supplement to figure 1. Similarly, Table 1 should also include values for non-sexual mycelia.

Images were added as suggested to Figure 1. Numbers of spectra found in vegetative mycelia were added to Table 1 and to the Table legend.

The use of the term "accurate" for sexual development is a little misleading as it implies that something is inherently wrong (i.e., "inaccurate"). It appears that while the number and size of cleistothecia are impacted by intermediates in this pathway, it does seem that this structure, and the ascospores are produced. I agree that these changes may be important, but I think the language surrounding this discussion needs to be more nuanced to better reflect the possible ecological impact (or lack thereof) of these differences.

We removed the term “accurate sexual development” and changed the heading of the paragraph to “Accumulation of anthraquinone intermediates in *mdp*/*xpt* mutants impairs sexual development” (lines 280-281). Further, we have rewritten the parts about the ecological impact (lines 317-321 and 565-568).

Even when appropriate complementation controls are used, there can be concerns about the use of mutants for ecological inferences that touch on fitness. In particular, some associations between secondary metabolism and fungal fitness that have been elucidated with mutants seem to have no reflection on natural patterns. For example, using mutants it has been shown that conidiation is decreased when aflatoxin production is knocked out. However, studies have not been able to find any evidence that non-aflatoxigenic isolates of Aspergillus spp. produce fewer conidia (even when comparing closely related isolates). The transformation process alone (without disrupting genes) can result in shifts in fitness (https://doi.org/10.1094/Phyto-80-1166) – not to mention off-target impacts of gene deletion. It is thus of great importance to consider how the mutations induced in this study might reflect mutations that occur and or persist in nature. Problematically, the WT does not accumulate many of the compound intermediates (figure S10) that are determined to inhibit the growth of other fungi, raising questions about the relevance of these experiments to nature. While it is fine to look at the impact of these intermediates, the authors must acknowledge the fact that these intermediates may not occur in nature and that isolates that do produce them may succumb to selective pressures that could be associated with their findings of aberrant sexual development (some clarification on how the authors can be sure that the aberrant cleistothetical development is bad would benefit the – I don't know that I disagree, but it is worth discussing).

We have included this important point that the wildtype does not accumulate the intermediates of the cluster and that therefore it is unlikely that they have an ecological function (lines 319-321) and in discussion (565-568).

Interpretation of the fungal-competition experiments is further complicated by questions that remain about the concentrations of pure/crude compounds used. Were concentrations of extracts at all reflective of what might be produced/secreted by *A. nidulans*? Indeed – are these compounds secreted at all? How does the 75 ug per disc reflect concentrations actually produced by the fungus? Can anything resembling inhibitory concentrations be produced on non-synthetic media that may be encountered in nature? Importantly if these compounds are not secreted, might they instead protect against the growth of other fungi directly on cleistothecia?

We performed additional experiments which revealed that the identified metabolites can be secreted to the medium. This is added to the results part (Line 198-202, Figure 2—figure supplement 4), and the methods are described (lines 776-780). It remains to be shown during what conditions the fungus might secrete these compounds in its natural habitats to protect cleistothecia and mycelia. The tested *A. nidulans* wildtype AGB552 does not produce emodin under tested laboratory conditions and therefore an ecological function is unlikely, but for the *A. nidulans* wildtype A4, emodin production was shown before (Bayram et al., 2016; lines 565-568).

75 µg per disc corresponds to the amount which was received from 6-7 fungal colonies. We included this information in lines 394-395 and 819-820.

I commend the authors for their use of three diverse fungivores. However, the simple food-choice test that was conducted does not provide much insight into the ecological role of xanthones. It is important to note that just because insects spend more time near colonies unable to produce mdp/xpt metabolites does not necessarily mean that these colonies are eaten more or have a fitness disadvantage. Other studies approaching similar questions, including those cited by the authors, have performed far more comprehensive assays to draw similar conclusions as the authors do here. For example, Cary et al. (2014) performed similar food-choice tests of fungal tissues to fungivores. However, in addition to offering fungivores a choice between two genotypes of fungus, they also performed no-choice experiments that help to elucidate the nature of antifeedant activity. Furthermore, Cary et al. assessed the weight of fungal structures in question, thus adding a fitness measure to their food-choice. It is also not uncommon for studies to also assess the toxicity of compounds of interest on insects.

We addressed this point by performing additional experiments with *Folsomia candida* springtail interactions with *A. nidulans* strains, which are summarized in the novel Figure 8.

We could show that the metabolites of the wildtype protect the cleistothecia itself as well as the sexual mycelium. We could also show that this effect does not correlate with a toxic effect of the metabolites for the predators. (Lines 433-455, Figure 8, Figure 8—figure supplement 3, method part lines 854-896).

The authors approach to the insect experiments is a little confusing in light of their hypothesis. It is my understanding of the authors argument that xanthones are targeted through Hulle cells to protect cleistothecia. However, the authors do not investigate if there is any difference in the protection conferred specifically to these structures. If xanthones do indeed have strong antifeedant effects, then it would be important to clarify if these effects are conferred preferentially to the cleistothecia or if they are realized by the entire thallus.

This point was addressed by our new experiments, which revealed that the animals prefer feeding on mycelium as well as cleistothecia without the final *mdp*/*xpt* xanthones (lines 433-448; Figure 8, Figure 8—figure supplement 3).

The authors identify a correlation between species that produce Hülle cells and those that produce the *mdp*/*xpt* metabolites. While this is intriguing, it is not evidence that these other species also produce *mdp*/*xpt* metabolites in a Hülle-cell associated manner. The text should be more cautious in its interpretation (lines 479-485).

The text was changed for being more cautious as suggested (lines 536-538).

The discussion should acknowledge a large body of literature that now ties close connections between secondary metabolism and primary growth raising questions about the authors suggestion that SMs are not "directly involved in growth".

We have added the ties between primary and secondary metabolism already in the introduction (Lines 46-48).

I do not find the speculation about the Emodin target very convincing. Additionally, I do not find conclusions about the ecological role of emodin production convincing given that emodin does not seem to be accumulated by the WT and thus the ecological function of emodin production rests in its derivatives.

We have added this concern as suggested (lines 565-568).